# Digital immunoassay for biomarker concentration quantification using solid-state nanopores

Liqun He [1,2], Daniel R. Tessier [1,2], Kyle Briggs [1], Matthaios Tsangaris[1], Martin Charron [1], Erin M. McConnell [1], Dmytro Lomovtsev [1] & Vincent Tabard-Cossa [1✉]

Single-molecule counting is the most accurate and precise method for determining the concentration of a biomarker in solution and is leading to the emergence of digital diagnostic platforms enabling precision medicine. In principle, solid-state nanopores—fully electronic sensors with single-molecule sensitivity—are well suited to the task. Here we present a digital immunoassay scheme capable of reliably quantifying the concentration of a target protein in complex biofluids that overcomes specificity, sensitivity, and consistency challenges associated with the use of solid-state nanopores for protein sensing. This is achieved by employing easily-identifiable DNA nanostructures as proxies for the presence ("1") or absence ("0") of the target protein captured via a magnetic bead-based sandwich immunoassay. As a proof-of-concept, we demonstrate quantification of the concentration of thyroid-stimulating hormone from human serum samples down to the high femtomolar range. Further optimization to the method will push sensitivity and dynamic range, allowing for development of precision diagnostic tools compatible with point-of-care format.

[1] Department of Physics, University of Ottawa, Ottawa, Canada. [2] These authors contributed equally: Liqun He, Daniel R. Tessier. ✉email: tcossa@uOttawa.ca

The nanopore field has seen tremendous progress in the last two decades, from the direct detection of single molecules to the realization of nanopore sequencing of nucleic acids, and more recently, the fingerprinting of amino acids[1–10]. One of the many promises of nanopores formed in solid-state membranes is to one day impact in vitro diagnostic (IVD) medical devices through the rapid and sensitive quantification of specific disease biomarker molecules in a point-of-need format. Solid-state nanopores possess attributes that make them good candidates for this type of application, including: the ability to tailor pore size to suit the target of interest; robust supporting membranes suitable for a range of operating conditions; and ease of integration with microfluidics and electronics in dense arrays. Unfortunately, progress towards diagnostics applications has been slow, due to a host of challenges. For example, many clinically relevant biomolecules are proteins, which are not always compatible with the high-salt-concentration conditions that provide optimal signal-to-noise ratio (SNR) in solid-state nanopore devices. Transport properties of proteins through the nanopore are complex and fast, resulting in challenging detection and analysis issues[11,12]. Solid-state nanopores in their native state lack specificity to recognize a protein of interest without additional functionalization[13–16] and are prone to clogging and producing a high rate of false positives, particularly in complex biological fluids like serum that contain many different proteins. Finally, even when fabricated with sub-nanometer precision[17,18], solid-state nanopores exhibit pore-to-pore variability in their capture characteristics and transport properties induced by minute geometric and surface charge variations which are not currently controllable, making generalization of results between nanopores a significant analysis challenge[19,20]. This reproducibility issue is a bottleneck for the development of technologies based on solid-state nanopores that often goes unaddressed.

Despite these challenges, a handful of recent studies have attempted to develop solid-state nanopore-based methods for specific target protein detection[21–28]. Keyser and coworkers first showed how DNA nanotechnology can be used to create long linear dsDNA carriers that contain short protrusions as barcodes that hold receptors to carry specific proteins, thereby enabling multiplexed digital detection[22,29]. Similar DNA nanocarrier schemes were employed by the groups of Edel and Ivanov to detect proteins and antibodies[25,26] from serum, while Morin et al[7]. used a modified peptide nucleic acid molecule[30,31] that sequence-specifically inserts into the DNA scaffold to form a triple strand, and that can bind to a target antibody. While promising, these nanocarrier approaches have a sensitivity that is fundamentally limited by the affinity of the receptor (antibody or aptamer) to the protein of interest and will not be able to detect biomarkers in the fM range far below the typical dissociation constant for the interaction. In addition, the size of the target must generally be comparable or larger than that of the receptor to produce easily distinguishable signals between the target bound and unbound states in the absence of additional labeling, often restricting the choice of receptors to aptamers, usually with reduced affinity compared to antibodies. Furthermore, the high salt concentrations in which nanopores typically operate optimally must be balanced with the lower salt concentrations necessary to maintain a good affinity. Finally, the folding of long nanocarriers translocating the pore complicates analysis and can lead to ambiguous and false positive signals[27].

Alternatively, in an effort to push the limit of detection, Chuah et al. used nanoparticles decorated with antibodies specific to their target to intentionally clog a large array of nanopores that are also decorated with antibodies, resulting in irreversible clogs being representative of a positive signal[32]. These proof-of-concept results represent early nanopore-based challengers to traditional methods of protein detection.

Many diseases, such as hypo/hyperthyroidism[33], many forms of cancer[34,35], infectious diseases such as tuberculosis[36,37], neurodegenerative diseases such as Alzheimer's[38] and multiple sclerosis[39–41] and even traumatic brain injuries[42], are often heralded by the presence of low concentrations (~fM–pM) of specific protein biomarkers in blood samples. However, traditional analog ELISA equipped with intensity-based optical readout, is often insufficiently sensitive to quantify the relevant concentrations[43]. This is due in part to the affinity of antibodies, since the dissociation constants for the available antibodies are typically larger than the clinically relevant concentration of the target in blood samples, and in part to the inadequate SNR of standard optical readouts[42,44].

In order to reach the required level of sensitivity while preserving specificity, digital immunoassay schemes are being developed that are able to overcome the limits imposed by both analog optical readout and relatively weak antibody pairings[42,44]. Some of these schemes are based on the use of paramagnetic micron-sized beads decorated with hundreds of thousands of capture antibodies[45], effectively turning each bead into a capture antibody with a significantly higher on-rate than that of individual antibodies, but while the same off-rate is maintained[46,47]. In one method, this is followed by partitioning of beads into individually optically addressable microwells for digital readout of the fraction of beads with bound targets, overcoming analog error modes by transitioning to a digital scheme while still using optical detection[42,48]. These digital optical schemes have demonstrated impressive sensitivity down to the fM and even aM level for some exceptional antibody/antigen pairings[42,49], but could be ill-suited for point-of-care use or future integration into wearable sensors due to the large form factor and complex optics needed for fluorescent readout. Nanopore sensors with their intrinsically single-molecule resolution and purely electrical readout are viewed as an attractive alternative to optical detection schemes for digital diagnostics technologies that offer a demonstrated path towards miniaturization and portability[50]. In this work, we employ DNA nanostructures as the basis for a digital counting scheme, to realize a robust solid-state nanopore electrical detection method. Using the characteristic electrical profiles of our DNA nanostructures combined with traditional bead-based sandwich immunoassays, we demonstrate the ability to quantify concentrations of a target protein from a serum sample. The proposed assay overcomes the specificity, sensitivity, precision, and consistency challenges associated with solid-state nanopore sensors for protein sensing from complex biological samples while using a digital readout scheme to overcome systematic error sources otherwise present. It provides a method to consistently measure target protein concentrations extracted from human serum against much higher concentrations of non-specific background molecules from the fM to the nM range for a single nanopore sensor, in a format that is amenable to parallelization to increase sensitivity, precision, and speed while allowing for the possibility of miniaturization for point-of-care use.

We demonstrate the utility of the method by measuring the concentration of thyroid-stimulating hormone (TSH), a staple biomarker for hypo/hyperthyroidism[33]. TSH is frequently used as a proof-of-concept target in model research assays and in the design of biosensors with optical, electrochemical, or electrical readouts[51,52], since its clinically relevant concentration covers a large range and it is considered a good test case assay for new technologies since existing ELISA tests for TSH are particularly performant. The results presented here on TSH are generalizable to a framework that can be applied to any target protein/antibody pairing without a need to compromise on the nanopore sensing performance. To ensure consistency of results between many different solid-state nanopores we apply our recently reported

method of controlled counting, allowing for direct comparison and consistent calibration between different nanopores[19]. Our results, acquired on 15 nanopores, 200 experiments, and >10⁵ single-molecule events demonstrate that protein concentrations measured using this nanopore digital counting technique are accurate, precise, and robust, forming the basis for versatile biomarker quantification strategy.

## Results

**Digital assay design**. The quantification of specific protein concentrations from biological samples (e.g., serum) with solid-state nanopores requires precise, accurate, and robust electrical identification of these targets on a single-molecule (i.e., digital) basis from a complex background. To enable digital detection with solid-state nanopores for precise and consistent target concentration measurements, we designed a pair of double-stranded DNA (dsDNA) nanostructures in the shape of shooting stars that can be bound pairwise via a complementary single-stranded DNA (ssDNA) junction strand to form an easily distinguishable dumbbell-shaped nanostructure, as illustrated in Fig. 1. We refer to these shooting star-like DNA nanostructures as probes (P-1 and P-2) when they are unbound, and as dumbbells (DB) when they are linked by the junction strand.

P-1 and P-2 are composed of 12-arm dsDNA stars with 11 arms 25 bp in length, and a 12th arm consisting of either a 175 bp (P-1) or a 150 bp (P-2) linear dsDNA tail. At the end of this extended arm is a 25 nt ssDNA region that is complementary to half of the junction strand (Supplementary Fig. 1), which allows a P-1 and a P-2 probe to bind together to form a dumbbell-like structure (Fig. 1 and Supplementary Fig. 2). A description of probe assembly and purification can be found in Supplementary Notes 1 and 2.

Using the components identified above, our assay adapts a sandwich immunoassay scheme to enable digital detection with a nanopore, as depicted in Fig. 1. Briefly, we employed magnetic isolation to efficiently capture target proteins onto antibody-coated paramagnetic micron-sized beads[44] and to facilitate the necessary washes to remove background molecules and dissociate non-specifically bound complexes. Secondary detection antibodies bioconjugated with streptavidin, were then added to sandwich each target between the pair of high-affinity antibodies. After equilibration and washing, short pieces of 50 nt biotinylated ssDNA (the junction strand) were added, which bind to the detector antibody and label each target. These junction strands have a photocleavable linker inserted between the biotin and the oligonucleotide sequence. Following additional washing to remove the excess unbound junction strands, the full immunoassay mixture was exposed to UV light to cleave the junction strands from the beads and release them into the solution in proportion to the concentration of bound targets. The beads were magnetically immobilized, and the supernatant containing the cleaved junction strands was recovered. The recovered supernatant was incubated with known concentration of probes and then mixed with a high-concentration salt solution for nanopore sensing. A more detailed description of the assay steps and components is provided in the "Methods" section.

Following nanopore analysis, the translocations of unpaired probes were classified on a single-molecule basis as a "0", while the translocations of dumbbells are classified as a "1", thus converting the electrical nanopore signal into a digital count.

In this scheme, junction strands serve as proxies for the target proteins, and for a fixed probe concentration, the fraction of dumbbells formed can be calibrated to report on the original concentration of target proteins in a clinically relevant sample. As expected from controlled counting[19], the use of relative counts of each population eliminates the error from varying intra- and inter-nanopore properties, allowing for highly reproducible assay performance between different nanopores.

**Nanopore characterization of DNA nanostructures**. In previous work, we showed that short, multi-arm dsDNA stars produce

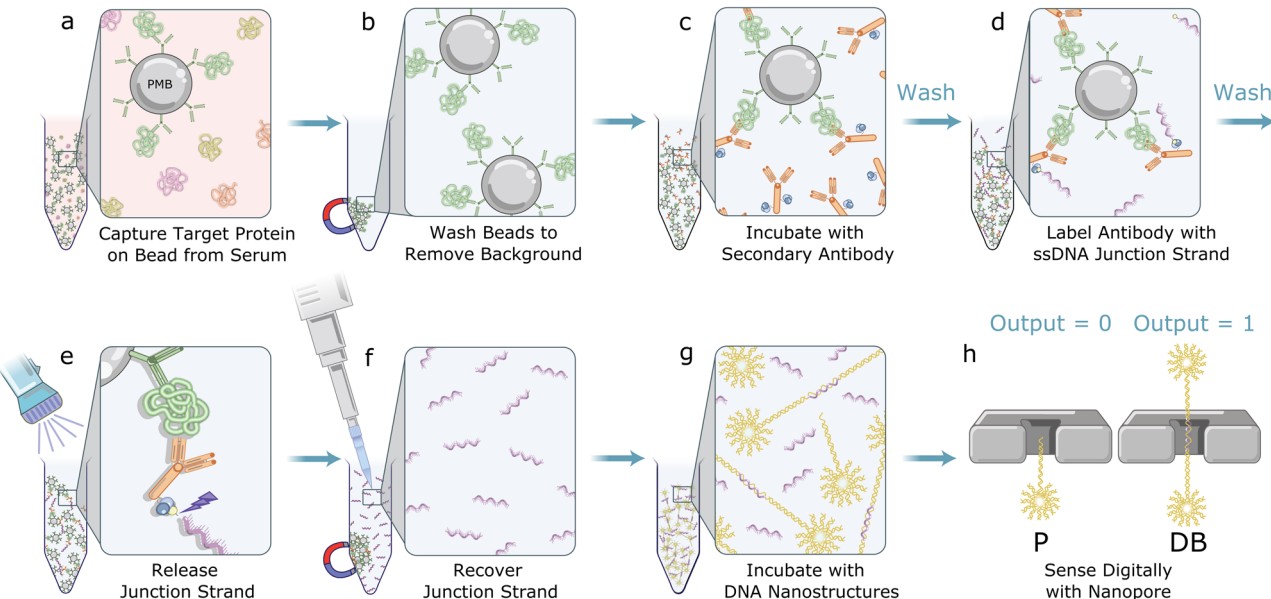

**Fig. 1 Schematic illustration of the digital immunoassay workflow with nanopore electrical detection. a** Paramagnetic beads (PMBs) conjugated with antibodies efficiently capture specific target protein in serum sample. **b** PMBs are pelleted and immobilized with a magnet and supernatant is removed to eliminate unbound molecules. **c** PMBs are resuspended and incubated with secondary antibody conjugated with streptavidin. **d** Following a wash, the immuno-sandwich structure is incubated with biotinylated ssDNA junction strand. **e** Following another wash, the solution is exposed to UV light to release the junction strand. **f** PMBs are pelleted and immobilized with a magnet and the supernatant containing the junction strand is recovered with a pipette. **g** Shooting star-like DNA probes are added to the solution containing recovered junction strand leading to assembly of a dumbbell-like DNA nanostructure. **h** Digital nanopore sensing to determine the fraction of probes to dumbbells.

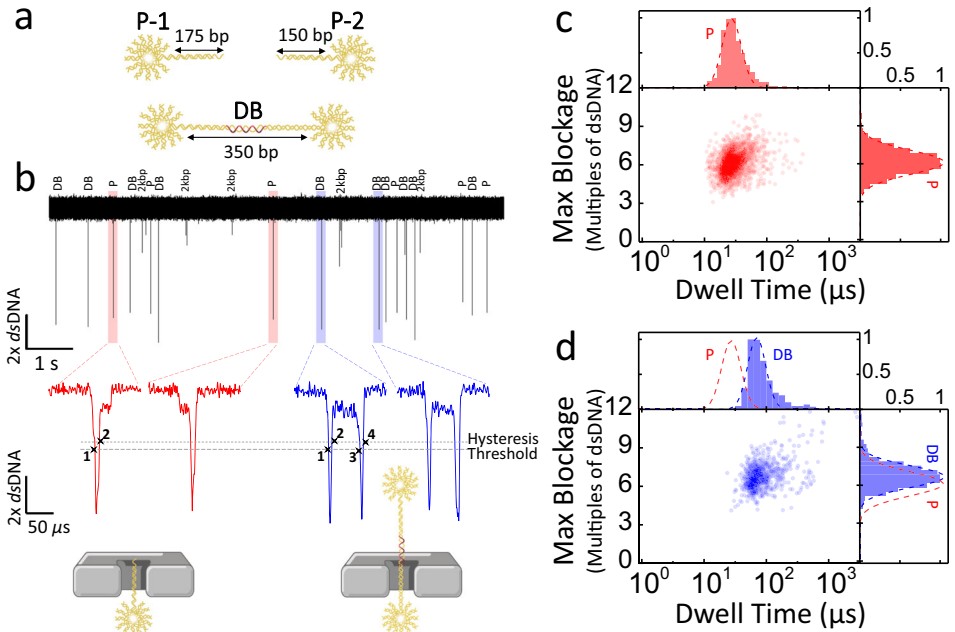

**Fig. 2 Nanopore translocation characteristics of the shooting stars probes and the dumbbells. a** Artistic representation of the shooting star probes (P-1 and P-2) and the dumbbell (DB). **b** 10 s current trace of a mixed population of both DNA nanostructures and 2 kbp dsDNA calibrator, with representative current traces showing individual translocation events corresponding to probes (left, red) and dumbbells (right, blue). Scatter plots and histograms of maximum blockage versus dwell time of the shooting star probes (**c**) and dumbbell (**d**). The fit to the probe distribution (P, red dash line) is overlayed with the dumbbell distribution to facilitate comparison between the two populations. Experiments are performed on an 11.5 nm pore in 3.2 M LiCl pH 8 with an applied bias of 100 mV. Displayed current traces are low-pass Bessel filtered at 500 kHz. Source data are available as a Source Data file.

robust and easily identifiable signals[53–58]. Here, we further modified the star nanostructures, extending one of the arms to form a linkage for the tail section of the probe, and added an internal carbon spacer in the middle of each star oligo's sequence to relax the otherwise highly charged and sterically stressed core of the 12-arm star structure, to help facilitate translocation through nanopores. The resulting nanostructure provides a characteristic electrical signature when translocating a nanopore (Fig. 2). Differences in nanopore sensing profiles between the probes P-1 and P-2, due to 25 bp difference in the length of the tail, were not distinguishable.

Figure 2 shows the nanopore translocation characteristics of probes and dumbbells. As expected, Fig. 2b shows the most commonly observed nanopore signal of the probes, which involves a deep blockage level 6x deeper than dsDNA alone, corresponding to the body of the star[53]. We normalized the nanopore current signal by the blockage produced by the unfolded translocation of 2 kbp linear dsDNA to remove the effects of any variations in pore geometry and operating conditions between experiments[53], thus facilitating comparison between experiments on different pores as detailed in Supplementary Note 3. These deep, 6x dsDNA, blockages are often (>90%) followed by a shallower 1x dsDNA blockage level corresponding to the tail, though bandwidth limitations sometimes (<10%) preclude resolving the tail part of the event. The mean translocation time was 27 ± 2 μs, though probe events do occasionally approach the 200 kHz bandwidth limit of the digital low-pass filter applied during the analysis. While this bandwidth limitation might in other contexts be problematic, the depth of the blockages ensures sufficient SNR to identify the event even if the signal is somewhat attenuated[59,60]. In contrast, Fig. 2b shows that when the dumbbells translocate, they produce two deep blockages separated by a shallower blockage level indicative of the linear dsDNA section between the two 12-arm DNA stars, with a mean passage time of 70 ± 2 μs. We observed a slight shift in

maximum blockage level (Fig. 2c, d) between the probes and the dumbbells since each dumbbell event produces two deep blockage levels and there is therefore a higher probability that at least one will be well-resolved, and the full blockage level correctly fitted. The presence or absence of this second, deep blockage level is confirmation of the presence of a junction strand in the sample.

In order to distinguish event types, we employed a simple threshold-crossing scheme detailed in Supplementary Note 3. Briefly, we counted the number of times the current trace in the event crossed a set of thresholds indicative of the transition between 1x dsDNA and 6x dsDNA blockage levels. With the threshold set to 2.5x dsDNA, a shooting star event registers 2 threshold-crossings, while a dumbbell event registers 4 threshold-crossings, as shown in Fig. 2b and Supplementary Fig. 4. The analysis of translocation events of the probes and dumbbells separately show that while the assembly into a dumbbell has only a mild effect on the passage time, the shape of the events are easily distinguishable. When using purified probes in the absence of junction strands we observed a false positive rate of <2%, primarily driven by either analysis artefacts or the binding of misassembled probe pairs (Supplementary Figs. 5 and 10). The latter is possible, for example, if both the probes P-1 and P-2 are missing the last two staple strands on the tip of their tails, in which case the probe pair overhang sequences are complementary and could form the dumbbell structure with a slightly shorter linkage in the absence of the junction strand. Future work will aim at resolving this by either having unique sequences for each staple in the entire tail on the respective probes, by increasing binding strength of the tail staples using longer sequences, and/or modifying the design of the DNA nanostructure labels.

**Nanopore digital response characterization.** To investigate and validate the dose response of our digital assay, we first characterized the response of the nanopore sensor as a function of different mixtures of known concentrations of junction strand

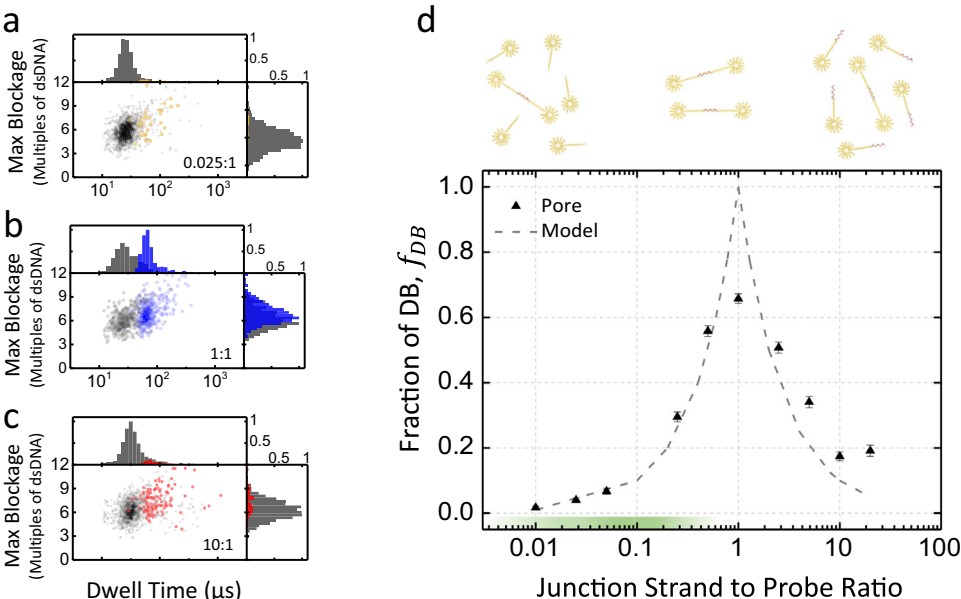

**Fig. 3 Dose response of the junction strand-to-shooting star probes ratio from 0.01:1 to 20:1. a–c** Scatter plots and histograms of maximum blockage versus dwell time (log scale), junction strand concentrations shown a) 500 pM (ratio 0.025:1); **b** 20 nM (ratio 1:1); and **c** 200 nM (ratio 10:1). Shooting star probes P-1 and P-2 are fixed at 20 nM in all three cases. Dumbbell events called using the thresholding algorithm are shown in color, while probe events are shown in gray. **d** Linear dose response on a log scale for junction strand concentration ranging from 200 pM to 400 nM and shooting star probes fixed at 20 nM, with ~1100 single-molecule events at each concentration. Dashed line depicts prediction from model of binding kinetics. The optimal operating range (0.01–0.5) is highlighted in green on the x-axis. Experiments are performed in 3.2 M LiCl pH 8, at 100 mV using a 12 nm pore, with the analysis threshold set to 2.5x dsDNA all experiments are low-pass Bessel filtered at 200 kHz for analysis. Error bars are calculated as described in Eq. 3. Source data are available as a Source Data file.

and probes. For this, we fixed the concentration of both probes P-1 and P-2 at 20 nM and varied the concentration of the junction strand, from 0.2 nM (ratio of 0.01:1, junction strand-to-probe pair) to 400 nM (20:1). Figure 3a–c shows the scatter plots of the maximum blockage depth versus dwell time for all single-molecule events recorded for three junction strand-to-probe pair ratios (0.025:1, 1:1, and 10:1). The corresponding scatter plots for all concentrations are shown in Supplementary Fig. 9. As expected, with increasing junction strand concentration below the fixed probe pair concentration, we observed a linear increasing fraction of events attributed to the passage of dumbbell nanostructures, reaching a maximum at a ratio of 1, before the relative number of dumbbells linearly decreased again. This linear dose response is plotted in semi-log scale in Fig. 3d.

To understand this non-monotonic response, we developed a simple computational model that assumes irreversible first-order binding kinetics of junction strands to probes. This model predicts that the fraction of dumbbells formed at equilibrium will be in proportion to the ratio of junction strands-to-probe pairs when there are fewer junction strands than probe pairs, and the inverse ratio in the opposite case, that is,

$$f_{DB}(x) = \min(x, x^{-1}) \qquad (1)$$

where $f_{DB}$ denotes the fraction of dumbbell events and $x$ the ratio of junction strand (proxy for protein) concentration ($c_{Js}$)-to-each shooting star probe pair concentration ($c_{P1} = c_{P2}$), that is, $x = \frac{c_{Js}}{c_{P1}} = \frac{c_{Js}}{c_{P2}}$.

This linear behavior can be understood readily by a simple intuitive argument. Assuming irreversible binding, if there are more probe pairs than junction strands ($x<1$), every junction strand that binds one of the probes will eventually be able to find another of the matching pair with which to bind, leading to one dumbbell per junction strand (i.e., $f_{DB} = x$), depicted in Fig. 3d as the green section of the junction strand-to-probe ratio range. On

the other hand, if there are more junction strands than probes ($x > 1$), probes will get capped and be unable to find a binding partner with a free binding site, with the probability of capping occurring before binding a partner being in proportion to the ratio of concentrations in the first-order approximation that diffusion times are not rate-limiting. This is in reasonably close agreement with our experimental results when $x \neq 1$, as can be seen in Fig. 3d. In this regime, the model predicts a linear increase in probe capping or linear decrease in the dumbbell fraction as we increase the junction concentration (i.e., $f_{DB} = x^{-1}$). The experimental data suggest that there is a limit at which the dumbbell fraction still occurs, nearing 20%, even in the presence of overwhelming numbers of junction strands. Because of this limit the range of excess junction strands to probes should be avoided for quantification. While the expected peak at $x = 1$ is present in the experimental data, in contrast to the prediction of our computational model, the experimental data show that the dumbbell fraction only reaches a plateau of about 60%. During assembly of dumbbells, particularly as concentration ratios near parity, it was noted that as expected from hybridization kinetics[61], the incubation times required to reach binding saturation are quite long due to progressive depletion of binding species as the reaction progresses. However, this underperformance of binding near the peak cannot be explained by short incubation times, since the reaction would have run past completion well before the experiments in Fig. 3d were conducted. To validate this dose response, we have also the same samples characterized by gel electrophoresis, which show good agreement with the nanopore results (see Supplementary Fig. 6).

The more likely explanation is that a fraction of the shooting star probes in our purified stocks were or became misassembled while stored, either prior to use or during manipulation ahead of nanopore sensing. This is a common issue with multi-component DNA nanostructures[53] which introduces potential false negatives

in the analysis, as discussed in more detail in Supplementary Note 3. To support the hypothesis that our DNA nanostructure can incur partial disassembly post purification, we characterized samples of fully assembled dumbbells that were purified by gel band extraction. The data are shown in Supplementary Fig. 7 and reveal that up to 20% of the single-molecule events do not generate the electrical signature of four threshold crossings expected for the translocation of the dumbbell (Fig. 2b). While further investigation is required to better understand the stability of these DNA nanostructures and determine their shelf life and optimal storage conditions, the presence of misassembled nanostructures is controlled by performing a calibration curve and serum sample testing with a single batch of probes.

Proper performance of the digital assay also requires setting the junction strand-to-probe ratios between 0.01 and 0.5 to maintain the dumbbell assembly in the optimal range (linear dose response). For junction strand-to-probe ratios well below 1 ($x \leq 0.5$), the assay should perform as expected even in the presence of up to 20% of mis-assembly, but nearing ratios of 1 ($x > 0.5$) this issue becomes limiting and results in a reduced precision. It is therefore important, both from a timing and accuracy perspective, to ensure that the concentration of shooting star probes chosen be such that the ratio of junction strands-to-probes is below 0.5. This is also where the hybridization reaction is fastest[61]. The lower limit of ~0.01 is currently set by the presence of false positives at a rate of <2%, though we hope to push this limit down by improving the DNA nanostructure design. This 0.01–0.5 regime is highlighted in green in Fig. 3d and defines the dynamic range for a given probe concentration. We expect that the current ~50-fold dynamic range of this assay scheme can be extended to ~2–3-log by improving the DNA nanostructure labels, or to an arbitrarily wide range of clinically relevant biomarker concentrations by incubating the unknown concentration of junction strands (proxies for protein) with different fixed concentrations of probes in parallel. For example, splitting the sample volume and testing it against three different probe concentrations could, in principle, provide a 5-log dynamic range. Likewise, the sensitivity of this assay, currently limited by the lower bound of the concentration ratio of junction strand-to-probe, $x = \frac{c_{\text{js}}}{c_{\text{p}}} \approx 0.01$, can be adjusted to accommodate any desired concentration range at the cost of linearly increasing the detection time of a single nanopore sensor as concentrations are reduced, though this increase in counting time can be offset through parallelization with an array of pores and other strategies that increase nanopore capture rate[62]. The limit of detection, while fixed by the parameters of a particular assay and the choice of probe concentration, can therefore be controlled by the counting time of the nanopore and for a fixed measurement time can be improved by speeding up the detection through parallelization, amplification, preconcentration, or capture rate enhancement schemes[19,29,53,62–68].

**Nanopore digital immunoassay for TSH**. To reliably quantify the concentration of TSH in human sera, our proof-of-concept protein target, we performed calibration curves using the full assay workflow presented in Fig. 1 and validated the reproducibility of the assay by testing the inter-pore variability. A five-point calibration curve was constructed using known concentrations of recombinant TSH (rTSH) (0.00, 0.15, 0.30, 0.60, and 1.2 nM) suspended in sample diluent (Fig. 4a). Note that after release of the junction strand (proxy for protein, Fig. 1f), a final concentration step was applied to the supernatant, increasing the concentration by ~17-fold before incubating for dumbbell assembly with a fixed 20 nM probe concentration (Fig. 1g). This particular probe concentration was selected in order to operate

the assay in the optimal range of junction strand-to-probe ratio previously discussed for Fig. 3d, and to limit the single nanopore recording time to minutes to count a statistically significant number of single molecules.

Figure 4a presents the analyzed nanopore data, showing the fraction of dumbbell formed, $f_{\text{DB}}$, as a function of the initial spiked rTSH concentration in each of the calibrators. The calibrators were run on three different nanopores: a 10 nm (pore 1, magenta diamonds), an 11.5 nm (pore 2, blue triangles), and a 12 nm (pore 3, cyan circles). The calibration curve for the fraction of dumbbell events ranges on average from ~3% (blank) to ~40% (1.2 nM). The calibrators exhibit a linear trend as expected and overlap within their error bar for all three pores, highlighting the pore-to-pore reproducibility of our digital immunoassay scheme for pores sizes in this range. The straight line shown in Fig. 4a is a linear fit to the calibration points for pore 3. For the different pores tested, the blank sample shows a background level varying from 2.5% to 3.4%. We had previously observed a <2% false positive rate from dumbbells formed from purified probes in the absence of junction strand (see Supplementary Fig. 10), which we attributed to analysis artefacts and the agglomeration of misassembled probe pairs. We therefore assign this additional ~1% to the values of the blank due to non-specific binding during the immunoassay, most likely of the secondary antibody to the beads not removed during washes. Limits of detection (LoD) were determined for each calibration curve at 2.5 standard deviations (s.d.) above background (blank) as commonly done[42]. LoD determined from the three pore runs were averaged for an overall mean LoD of ~20 pM for these particular choice of assay workflow and parameters.

The gray squares in Fig. 4a represent the idealized values of the five rTSH calibrators for junction strand-to-probe ratios of 0:1, 0.125:1, 0.25:1, 0.5:1, and 1:1 as interpolated from Fig. 3d. These interpolated values of the ratios assume no losses due to disassociation of the components from the immunoassay during the wash steps, nor of reagents from sticking to tubes walls, and a 1:1 labeling of the detection antibodies with junction strands (i.e., a perfect translation of each protein target to exactly one junction strand). Since losses are to be expected throughout the assay[69], we empirically estimate these losses by comparing the interpolated values to our experimentally observed values. The difference in dumbbell fraction seen in Fig. 4a indicates that we are on average recovering half as many junction strands as there are target proteins present.

Next, we investigated the accuracy of the nanopore digital signal with clinically relevant biological samples and validated the reproducibility for protein concentration quantification in these conditions. To accomplish this, we spiked 0.48 nM of rTSH into a human serum sample, with a predetermined undetectable level of TSH (see "Methods" section) and measured it on the three different pores as above. Linear fits to the calibration points for each pore are used as calibration curves to convert the observed fraction of dumbbell into the concentration of TSH originally present in the human serum sample (i.e., $C_{\text{protein}} = f_{\text{DB}}/\text{slope}_{\text{cal.}}$). Figure 4b shows that the observed fraction of dumbbell formed are 19 ± 2%, 19 ± 2%, and 19.2 ± 0.9% corresponding to 0.54 ± 0.04, 0.61 ± 0.05, and 0.59 ± 0.03 nM of rTSH, respectively. These triplicate measurements are in good agreement with one another. Interestingly, the quantified concentrations of TSH are systematically higher than the spiked concentration of 0.48 nM. The spike recovery analysis shows that we have a recovery of 121 ± 5%, which is close to the acceptable range of 80–120%[70]. We attribute discrepancies here to non-specific binding and matrix effects in the presence of serum since the calibration curves were constructed from spiked rTSH in sample diluent. In

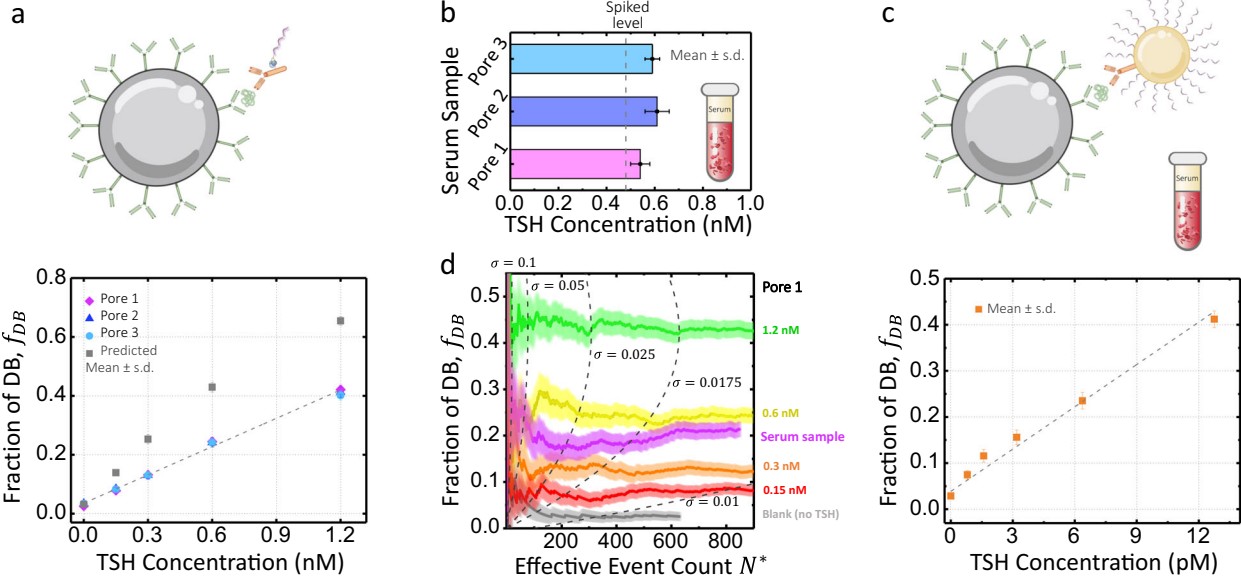

**Fig. 4 TSH assay calibration curve and TSH serum sample concentration quantification using solid-state nanopore digital detection. a** TSH calibration curve concentration, 0 (blank), 0.15, 0.3, 0.6, and 1.2 nM, repeated on pore 1 (10 nm, magenta diamonds, $N = 1251$, 1247, 1750, 1301, and 1239 single-molecule events), pore 2 (11.5 nm, blue triangles, $N = 1120$, 1911, 1286, 1097, and 1103 single-molecule events), pore 3 (12 nm, cyan circles, $N = 1817$, 1736, 2079, 2699, and 2140 single-molecule events), and fractions interpolated from data in Fig. 3d (gray squares). The error bars represent one standard deviation (S.D.) following Eq. 3. **b** 0.48 nM TSH-spiked serum as measured by the assay on the same three pores, with $N = 1522$, 1602, 1103 single-molecule events. **c** Results of assay with the gold nanoparticle amplification scheme for TSH spiked in serum at 0.8, 1.6, 3.2, 6.4, and 12.8 pM on a 12.2 nm pore (orange squares, $N = 1785$, 1524, 980, 1177, 865, and 945 single-molecule events). **d** Fraction of dumbbell events as a function of the effective event count $N^* = N_{DB} + N_P/2$ detected for dumbbells and probes on pore 1. The colored bands represent one s.d. and the dashed lines represent the values of $N^*$ needed for a given dumbbell fraction $f_{DB}$ to have $\sigma_f = 0.1$, 0.05, 0.025, 0.0175, and 0.01. The probes P-1 and P-2 are fixed at 20 nM for 1:1 assay and 15 nM for the amplification assay. Experiments are performed in 3.2 M LiCl in a mixture with 2 kbp linear dsDNA at 100 mV and an intra-crossings threshold of 2.5x dsDNA, all experiments are low-pass Bessel filtered at 200 kHz for analysis. Source data are available as a Source Data file.

future work, we anticipate that matrix effects can be normalized by running the calibration curve in TSH-depleted serum or could be reduced by further increasing the dilution factor of the sample from 4× to ≥10×[70]. Additional data sets of calibration curves and serum measurements on two other pores using a different reagent batch yielded similar results as shown in Supplementary Fig. 11. Nanopore assay results were further validated by gel electrophoresis (see Supplementary Figs. 12 and 13), and five additional measurements of rTSH in human serum sample from 0.15 to 1.2 nM are shown in Supplementary Fig. 14.

To further improve the sensitivity of our digital immunoassay and lower the LoD for a fixed nanopore counting time of a few minutes, we implemented an amplification scheme to detect rTSH down to the high femtomolar range. To achieve a ~100× amplification, we replaced the biotinylated ssDNA junction strand with a detection complex composed of 30 nm gold nanoparticles decorated with detector antibodies and hundreds of junction strands following protocols by Mirkin & Co[45,71]. This effectively translates one protein into hundreds of ssDNA junction strands instead of a single one as illustrated in Fig. 4c. Amplification experiments were carried out using the same assay workflow. This time, we measured concentrations of spiked rTSH in human serum samples at 0.8, 1.6, 3.2, 6.4, and 12.8 pM. Figure 4d shows the detected fraction of dumbbell formed of 7.5 ± 0.9%, 12 ± 1%, 16 ± 2%, 24 ± 2%, and 41 ± 2%, with a blank serum measured at 2.9 ± 0.6%. LoD for the particular parameters of this amplified digital assay is calculated to be 385 fM. Additionally, we performed a homebrew ELISA employing the same workflow and reagents as our nanopore digital assay to validate the performance of our assay components with an optical readout (see Supplementary Fig. 15).

Since the speed at which single molecules are counted is dictated by the capture rate of the nanopore (a Poisson process)[19], the evolution of dumbbell fraction as a function of cumulative number of events offers insights into the assay time needed to accurately identify a particular concentration with a desired level of precision. Standard error propagation gives the mean and s.d. of the dumbbell fraction $f_{DB} \pm \sigma_f$ as:

$$f_{DB} = \frac{N_{DB}}{N^*} \qquad (2)$$

$$\sigma_f = \sqrt{\frac{f_{DB}\left(1 - f_{DB}\right)\left(1 - \frac{f_{DB}}{2}\right)}{N^*}} \qquad (3)$$

$$N^* = N_{DB} + \frac{N_P}{2} \qquad (4)$$

where $N_{DB}$ is the number of "1" events (dumbbells) and $N_P$ is the number of "0" events (probes), assuming Poisson error on both $N_{DB}$ and $N_P$, i.e. that the s.d. of each is equal to its square root. $N^*$ is the effective event count. Note the factor of 2 is included to pair probe events for the sake of comparison to the dumbbell fraction.

Like the standard error of a Poisson process or the s.d. of a capture rate, the uncertainty in the ratio of two Poisson processes scales inversely with the square root of the total event count. In the limit of $f_{DB}$ approaching either extreme of 0 or 1, where one event type dominates the counting, the numerator of Eq. 3 approaches zero, and the uncertainty on $f_{DB}$ is small and relatively insensitive to total event count as a result. Although intermediate values of $f_{DB}$ show higher uncertainty for a given event count, $\sigma_f$ can be reduced greatly by recording more events.

The effective event count, $N^*$, required to determine the fraction of events which are counted as "1" (here dumbbells) accurately within a predetermined error of $\pm\sigma_f$ depends on the value of the dumbbell fraction $f_{DB}$ itself. Figure 4d shows the evolution of the multiple measured dumbbell fractions of pore 1, as a function of the effective event count with respective error bars plotted following Eq. 3. Figure 4d shows that at least 129, 173, 262, and 307 effective event counts are required to obtain an absolute uncertainty of $\sigma_f = 0.025$ (2.5%), for TSH concentrations of 0.15, 0.3, 0.6, and 1.2 nM, respectively. The minimal event count needed to distinguish two different concentrations also depends on the respective values of $f_{DB}$[72,73].

Our discussion has thus far pertained only to the number of single-molecule translocations (counts) instead of measurement time[19]. Under our experimental conditions and with the TSH concentration ranges presented, the DNA nanostructures have a capture rate of ~1 Hz nM$^{-1}$ on a single 10 nm nanopore, resulting in sensing times on the order of tens of minutes required for precisions of $\sigma_f \approx 0.01$. Evidently, the time to response of the nanopore sensor can be reduced or the sensitivity and precision of the digital immunoassay increased by counting single molecules more rapidly in a fixed time. We have shown that the sensitivity of our digital nanopore assay can be significantly improved by amplification, yet in this configuration it is still limited by measurement time. As discussed previously, using a lower probe concentration can reduce the sensitivity by an equal factor at the cost of proportionally increasing the counting time unless the detection rate is accelerated by a combination of parallelization (array of pores), preconcentration, or capture rate enhancement schemes. See Supplementary Note 5 for an example of the latter strategy. In the idealized limit of rapid nanopore detection, the enhancement factor of the $K_D$ by the antibody-coated magnetic bead and non-specific interactions would ultimately become the limiting factors.

We have shown that solid-state nanopores can be used as precise and specific digital sensors for magnetic bead-based sandwich immunoassays, using DNA nanostructures as proxies for the presence or absence of a specific target protein. We were able to accurately and most importantly consistently quantify the concentration of spiked protein in human serum in the high fM to the low nM range, overcoming several of the major challenges associated with using solid-state nanopores to quantify the concentrations of biomolecular targets in complex biological samples. This represents a ~1000× improvement in sensitivity compared to the previous nanopore studies quantifying protein concentration from serum[25–27]. This digital scheme, based on the electrical counting of single-molecules, is an effective solution to the pore-to-pore consistency issue that has been slowing the development of bioassays on solid-state nanopores. The shooting star nanostructures used here give a high SNR of 20 (at 200 kHz), while using high recording bandwidth to better resolve translocation events. The advantages of the antibody-coated magnetic bead approach compared to DNA nanocarrier schemes is that the affinity (more precisely the on-rate, $k_{on}$) of the magnetic bead–antibody complex to capture protein targets is enhanced by a factor equal to the number of antibodies present on the bead surface, essentially turning each bead into a much more efficient antibody and enabling fM sensitivity[74]. The selectivity is also increased by washing away background molecules and non-specifically bound objects, leaving only molecules of interest to translocate through the pore, thereby reducing false positives.

The next foci for performance optimization for our proposed assay are to increase the dynamic range and to further decrease the limit of detection while reducing the assay time to <1 h. Both dynamic range and limit of detection can, in principle, be arbitrarily extended by varying the probe concentration but are practically limited here by the measurement time. As discussed previously, the current optimal dynamic range spans a ratio of junction strand (proxy for protein)-to-probe of ~0.01 to ~0.5, and for probe concentrations below <10 nM the incubation time required to hybridize the junction strand to two nanostructured probes to assemble the dumbbells becomes much too long (days) to be of practical use[61]. To tackle this timescale problem, techniques to control and increase the rate of nucleic acid hybridization reactions, such as isotachophoresis[75], could be employed to extend the dynamic range from 10 nM down to the fM levels. Combined with the strategies for improving the rate of single-molecule counts by a nanopore already discussed (parallelization, amplification, preconcentration, and capture rate enhancement schemes), we expect the proposed solid-state nanopore-based digital immunoassays scheme to reach low fM levels with a ~3-log dynamic range to undertake relevant clinical applications.

## Methods

**Probe and dumbbell assembly.** Probes are composed of oligos purchased from Integrated DNA Technologies. Oligo sequence information is available in Supplementary Table 1. Probe 1 was assembled using the 12-star oligos and extension oligos A to G. Probe 2 was assembled using the 12-star oligos and extension oligos A to F′.

Equimolar concentrations (final concentration of 0.3 μM for each oligo in a total reaction volume of 360 μL) of all oligos were added in assembly buffer of 1x TAEMg (40 mM Tris, 20 mM acetic acid, 2 mM EDTA, and 12.5 mM magnesium acetate, pH 8), heated to 95 °C for 5 min, cooled to 90 °C, ramped down from 90 °C at a rate of 0.4 °C min$^{-1}$ to 60 °C, from 60 to 26°C at a rate of 0.03 °C min$^{-1}$, and snap cooled to 4 °C in MiniAmp Plus Thermal Cycler (ThermoFisher Scientific, #A37835). After assembly probes were visualized on 2% agarose gel in 0.5x TBE buffer, pH 8.2 (Fisher Scientific, BP1333-1). Generuler 1 kb plus DNA ladder (ThermoFisher Scientific, SM1331) was used as a reference guide for nanostructure migration. GelRed (Biotium, #41003) was used for visualization of the DNA bands on all agarose and PAGE gels.

Probe batches were divided and loaded in multiple lanes on four 5% Mini-PROTEAN TBE polyacrylamide pre-cast gels (BioRad, 4565013). The band of interest was excised, and gel extracted using a custom gel extraction apparatus. Purified probe concentrations were estimated using optical density measurements made on a Take3 micro-volume plate and EPOC 2 spectrophotometer (BioTek, BTEPOCH2).

In order to assemble dumbbells directly, probe concentration was fixed at 20 nM and an increasing concentration of junction strands was added, ranging from 0.2 nM (0.01:1 junction:probe) to 400 nM (20:1 junction:probe), incubated for 2 days at room temperature (RT, 22 °C) with a total volume of 35 μL in 1x TAEMg buffer.

**Assay components.** A step-by-step protocol describing the assay procedure can be found at Protocol Exchange[76]. Paramagnetic beads of 2.7 μm diameter were conjugated with mouse monoclonal anti-human TSH beta subunit capture antibody (Fitzgerald Industries International, 10C-CR2151M4). Conjugation was performed in accordance to SIMOA Homebrew Assay Development Kit procedures (Quanterix, 101354). Then, 0.3 mg/mL (28x dilution) TSH antibody was incubated with 1.4× 10$^9$ beads. A non-competing detection antibody–streptavidin conjugation was done with mouse anti-human TSH beta subunit detection antibody (Maine Biotechnology Services, MAB130P) and conjugated to streptavidin (1.6x dilution) using a Streptavidin Conjugation Kit (Abcam, ab102921).

For the assay calibration curve, 7.2× 10$^7$ bead-capture antibody conjugates were mixed with varying amounts of recombinant TSH (rTSH) (BiosPacific, J11030) at concentrations of 0, 0.15, 0.3, 0.6, and 1.2 nM in 1x sample/detector diluent (Quanterix, 101359) for a total volume of 500 μL (volume used for all assay steps unless otherwise noted) and incubated for 1 h at RT, 22 °C. To keep beads in suspension, tubes were placed on a 360° Multi-Functional Tube Rotator (VWR, PTR-35). All subsequent incubations and washes (>30 s) were performed on the rotator. For the 480 pM rTSH-spiked serum sample a similar mixture was applied but with a 4x diluted serum (125 μL). De-identified human serum samples were purchased from BioIVT and collected from consented donors under IRB-approved protocols following BioIVT standard operating procedures and tested for TSH on an in-house TSH assay (homebrew) using the SiMoA platform (HD-1 Analyzer, Quanterix). TSH levels in the samples used are below the lower limit of detection (LoD 0.18 fM), therefore undetectable in our nanopore assay. A 4x dilution was applied to the serum sample to reduce matrix effects[77]. For the amplification assay measurements, identical procedure was followed: 10$^7$ bead-capture antibody conjugates were mixed with varying amounts of recombinant TSH at concentrations of 0, 0.8, 1.6, 3.2, 6.4, and 1.2 pM in 4x diluted serum of total volume of 650 μL, and washes were done with 200 μL 1x wash buffer instead. After initial incubation, three wash steps were performed by magnetically immobilizing

the paramagnetic beads, removing the supernatant, and resuspending in 1x wash buffer 1 (Quanterix, 100486), with 5, 10, and 15 min intervals between each wash. Following washes, the immobilized beads are removed from the magnet and the pellet is resuspended in 500 µL 1x sample/detector diluent containing 6 nM of detection antibody–streptavidin conjugate and incubated for 1 h at RT. For the amplification assay, the gold nanoparticle amplification complex was added and incubated. After incubation, three 30 s washes with 1x wash buffer were performed to remove any unbound detection antibody. To complete the immunoassay sandwich for all samples, 12 nM of biotinylated ssDNA junction strand were added and incubated in 1x sample/detector diluent for 15 min at RT. To remove any excess of unbound junction strands, three 30 s washes using 1x wash buffer were performed, followed by buffer exchange to 1x TAEMg with 0.1% Tween-20. Supplementary Fig. 8 demonstrates that the presence of Tween-20 in the 1x TAEMg and 0.1% release/assembly buffer does not appear to affect nanopore background or assembly.

For the gold nanoparticle (AuNP) amplification complex, 30 nm standard gold nanoparticles (1 OD, Cytodiagnostics, G-30-20) were prepared following the protocols developed by Mirkin & Co[45,71] as well as manufacturer technical notes. First, 0.1 M borate buffer was added to a total of 1 mL of 30 nm stock AuNP to adjust pH to 9.0, followed by the addition of 4 µg of detection antibody (Maine Biotechnology Services, MAB130P). After a 30 min incubation, 25 µL of 30 µM reduced DNA oligos were added and incubated for 1 h. The solution was then salt-aged with 0.15 M NaCl over the course of 3 h at RT with gentle vortexing, and further incubated at 4 °C overnight. Then, 50 µL of 10% BSA was added to the solution and incubated for 20 min to passivate and stabilize the AuNP. The final solution was centrifuged at 4500g for 15 min, and the supernatant was removed. This wash step was repeated 3 times and final complex was resuspended in 200 µL of 1xPBS with 0.025% Tween 20.

All samples were exposed to UV using a 3 W LED flashlight (LIGHTFE, UV301D) at a distance of 1 cm for 20 min to cut the photocleavable linker present at the 5′ end of the junction strand and release it from the immuno-sandwich. For the amplification assay, the junction strands are released by an incubation of 45 min in 0.2 M DTT (Dithiothreitol, ThermoFisher Scientific, A39255) and 0.3 M NaH₂PO₄ buffer. Junction strands were recovered by magnetically immobilizing the remaining immuno-complex and recovering the supernatant with a pipette. To match the sensing range of the nanopore, a concentration step was performed to reduce the volume from 500 to 30 µL using Amicon Ultra-0.5 Centrifugal Filter Unit (Millipore Sigma, UFC500396). This yields an approximately 16.7x concentration increase of the junction strand into the low nM range for optimal nanopore sensing. Dumbbells were assembled simply by adding the probes to the junction mixture at a final concentration of 20 nM each (15 nM in case of the amplification assay) and incubating for 2 days. Note that shorter dumbbell assembly incubation times can be employed when the junction to probe ratio is far from 1 in either direction, however, 2 days was used to ensure complete assembly at the 1:1 ratio and avoid timing-related errors during proof-of-concept measurements. Further discussion regarding DB assembly time can be found in Supplementary Fig. 6).

**Nanopore fabrication.** Nanopores were fabricated in 12 nm thick free-standing SiNₓ membranes (Norcada, NBPX5004Z) using controlled breakdown (CBD)[17,78]. CBD was performed in 1 M KCl buffered with 10 mM HEPES at pH 8 and pores were grown to 9–12 nm in 3.6 M LiCl buffered with 10 mM HEPES at pH 8 using moderate voltage conditioning[78,79]. Prior to fabrication, the chips were cleaned using air plasma for 70 s and painted with a layer of PDMS to reduce high-frequency noise.

**Nanopore sensing.** The DNA nanostructures in 1x TAEMg pH 8 were mixed with a high molarity LiCl solution to reach a final concentration of 3.2 M LiCl for nanopore sensing, where 5 µL of the nanostructure solution (stock 20 nM or 15 nM) was added to 35 µL 3.6 M LiCl buffered with 10 mM HEPES at pH 8. Linear 2 kbp dsDNA fragments (ThermoFisher Scientific, SM1701) were always added to the mixture of nanostructure sample as a molecular ruler (with final concentration of 2.4 nM) to normalize away pore geometry variations during post-processing. Samples were introduced to the cis side of the chip and a negative voltage was applied to the cis side with the trans side grounded. The ionic current recordings were performed in MATLAB 2013a (32-bit) using the VC100 current amplifier (Chimera Instruments) with sampling frequency of 4.17 MHz and a bandwidth of 1 MHz and were subsequently software low-pass Bessel filtered as needed. The raw data files are accessible via Federated Research Data Repository: https://doi.org/10.20383/102.0483.

**Data analysis.** Nanopore signals were analyzed using a custom implementation of the CUSUM+ algorithm[55,56], which is freely available online[80] (https://www.github.com/shadowk29/CUSUM). A digital low-pass filter of 200 kHz was applied unless otherwise specified. Dumbbells events were distinguished using threshold crossing, with events containing exactly four threshold crossings identified as dumbbells, and exactly two threshold crossings as probes. Events with any other count of threshold crossings were rejected from analysis. Note that band-width limitations occasionally cause one pair of threshold crossings to be missed, resulting in a false negative. While improvements to analysis will be able to somewhat mitigate this issue going forward, in general, the proportion of events in this category are relatively consistent between pores and can be calibrated into the

measurement with minimal loss of precision and accuracy. Examples of all observed event types are shown in Supplementary Note 3. ImageJ was used for band intensity in gel images, OriginLab for plotting data.

**Reporting summary.** Further information on research design is available in the Nature Research Reporting Summary linked to this article.

## Data availability

Raw data supporting the immunoassay results within this paper are accessible via Federated Research Data Repository: https://doi.org/10.20383/102.0483. Any other data including validation and repeated experiments are available from the corresponding author upon reasonable request due to limited storage available. Source data are provided with this paper.

## Code availability

Nanopore data analysis was done using in-house program available at https://www.github.com/shadowk29/CUSUM

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

## Acknowledgements

The authors would like to thank John Pezacki, Mark Hayden, Craig Jeffrey, and Jeffrey Huff for fruitful discussions. The authors would like to acknowledge the support of the Natural Sciences and Engineering Research Council of Canada (NSERC), through funding from grant #CRDPJ 530554-18, and the National Institutes of Health, through funding from grant R01 EB031581.

## Author contributions

Adapted digital immunoassay scheme for nanopore readout: D.R.T., K.B. and V.T.C.; Designed proxy structures: D.R.T., K.B., L.H., M.C., M.T. and V.T.C.; Performed assay steps: D.R.T., L.H. and E.M.M.; Characterized and prepared proxy structures: L.H., D.R.T., M.T. and D.L.; Performed nanopore experiments: L.H.; Analyzed nanopore experiments: L.H., K.B. and M.C.; Assay design and biomolecular experimental protocols: D.R.T. and V.T.C.; Designed nanopore experimental protocols: L.H., K.B. and V.T.C. The manuscript was written through contributions of all authors. All authors have given approval to the final version of the manuscript.

## Competing interests

K.B. and V.T.C. are co-founders of Northern Nanopore Instruments and have an equity interest. All other Authors declare no competing interests.
