## [Peer Review File. · Nature Communications]

Reviewers' Comments:

Reviewer #1:

Remarks to the Author:

Tessier, He and coworkers developed an assay for biomarker concentration quantification. The assay is based on the capture of antibodies on beads that have a releasable short oligo. Release is done by (photo-)cleavage of the oligo and then assembly of a DNA nanostructure into dumbbells that is then transferred to 3.2M LiCl salt concentrations to allow for detection in solid-state nanopores. The authors translate their target protein signal into a DNA signal (junction strand) that then is detected. The concept of translating a protein into a DNA signal is not novel and was published before in Kong et al. *Adv. Funct. Mat.* 2019 <http://dx.doi.org/10.1002/adfm.201807555>. A paper that curiously is not cited. So the key idea to use nanopore sensing for the detection of proteins after translating the signal into a nanopore signal is not novel. The authors use a different type of DNA nanostructure to detect the presence of the junction strand by designing 12 arm shooting start structures. The paper is interesting for the nanopore sensing community as it shows another approach how DNA self assembly can be used to enable sensitive and specific sensing. However, the paper does not solve any problem that is relevant outside of the nanopore community as all scientific questions (dynamic range, false positive background etc.) are actually CAUSED by the chosen nanopore sensing system. The paper could be published in a journal like *Nanotechnology* or *ACS sensors*.

1) The authors nicely summarise the challenge of detecting low concentration targets on page 4, lines 74-77: "This is due in part to the affinity of antibodies, since the dissociation constants for the available antibodies are typically larger than the clinically relevant concentration of the target in blood samples, and in part to the inadequate SNR of standard optical readouts." The authors address neither of these key issues on their work. They rely on antibody binding to beads, introduce a nucleotide label that needs to be detected by a complicated process with nanopores. It is completely unclear why this could not be done with a quencher-dye, FRET or any other single-molecule optical assay as everything up and including the cleavage is not in any way relying on nanopore sensing. In case the authors want to make the paper relevant for the sensing community they need to compare their readouts with ELISA assays or the optical assays they are criticising. None of these things are presented in the manuscript.

2) The key figure of the paper 4b contains a single TSH concentration measured with three different nanopores with non (or barely) overlapping error bars. Before any publication the authors have to present data with at least two more concentrations from the serum sample. The data in Figure 4b is simply not enough.

3) The authors have to show that their method outperforms other detection systems like ELISA when the SAME process is used to translate the signal.

4) The authors claim that nanopores are more amenable to bedside/point-of-care applications. For the nanopore sensor this is true but do they suggest that they can perform their assay in Figure 1 near the bedside on a nanopore systems they have to show this.

5) In the outlook the authors make the claim that this system can reach fM detection levels for proteins when this is clearly limited by the antibody affinities as they discussed in point 1). They need to either qualify this claim or explain how they aim to overcome the issue of low affinity antibodies as the nanopore sensor does not solve the key issue of affinity.

Due to the above issues the manuscript cannot be accepted in *Nature Communications* in the current form.

Reviewer #2:

Remarks to the Author:

This work reported a digital nanopore counting approach to quantify the target protein concentration from complex biofluids. The general strategy relies on the magnetic bead-based

sandwich immunoassay to convert the protein numbers to an equivalent number of junction ssDNA strands, which is then quantified by using the shooting star probes to form a recognizable dumbbell-like DNA nanostructure. In my assessment, this is a well-designed study and is of significant interest to the research community to develop ultrasensitive immunosensors. Major concerns I have for this strategy are the limited dynamic range, the complexity of the assay, and the sensing time. Although the authors did acknowledge these issues in the manuscript in a conceptual framework, these issues should be addressed with experimental data. I would recommend a major revision of the current manuscript.

Scientific rigor

1. Page 8, line 176 mentioned "improving structural stability and reducing clogging in the nanopore". Authors should elaborate on why the internal carbon spacer can help reduce clogging. It's best to show experimental data to support this claim.
2. Figure 2b, what does the label "2x dsDNA" mean? Please clarify if the unit dsDNA refers to the current blockage of the linear 2 kbp dsDNAs. I suggest using a plain vanilla current level so that other researchers in the community can benchmark (even though the pore size variation could change this current level, but readers can at least have some idea if authors can show an example).
3. Figure 2c&d, the maximum blockage depths reported are normalized by the blockage of 2 kbp dsDNA. Since the 2kbp events were collected, it will be better to add it to the scatter plot in Figure 2c and 2d. what is 2kbp events' distribution. If using 2kbp dsDNA to normalize, what exact value should be used, the peak or the mean? In addition, what is the blockage shape for the 2kbp ruler? I suggest the authors show a representative shape for the 2kbp ruler.
4. Figure 2d, please clarify if the dashed line (annotated 'p') in Figure 2d represents a replot of Figure 2c. How is the dwell time defined for dumbbell structure? From point 1 to 4 in figure 2b?
5. Figure 3, ten cases of different junction strand to probe pair ratios (from 0.025:1 to 10:1) were performed. It will be more informative to readers if the authors plot all of the scattering plots and indicate the number of detected events for each case.
6. Regarding the model (Eq.1), What is the physics behind your model? Any references? I suggest the authors rationale why this model applies here.
7. Figure 4, please add the pores 1,2 and 3 size information to the caption.
8. Page 23, line 501 mentioned 'TSH levels in the samples used are below the lower limit of detection (LoD 0.18 fM), therefore undetectable in our nanopore assay'. What is the limit of detection in the "Nanopore Digital Immunoassay for TSH" section? I suggest adding the LoD analysis data.
9. In supplement: Figure S7c showed some faint bands above the purified dumbbell bright band, and those were also observed in Figure S6a (from ratio 0.01:1 to 2.5:1). What is the possible cause for these bands?
10. Based on the result in figure 3d, one should add a particular probe concentration to the system for the optimal range of junction strand to probe ratio. In a system with an unknown TSH concentration, how can one determine the probe concentration in the reaction to prevent junction to strand ratios higher than 1? Of course, as the author mentioned, testing the sample against different probe concentrations could, in principle, provide a 5-log dynamic range. Nevertheless, considering the complexity of this strategy, authors should comment on the cost (if run parallelly) or the time to detect (if run sequentially). What are the key benefits of such a system as compared to other commercialized digital systems (e.g., Quanterix system). I would think this is the major drawback of the proposed digital assay towards real-world applications. These issues should be addressed with experimental data.

Typos and format

1. Page 9, line 189: the dumbbells translocate is in Figure 2b, not "Figure 2d".
2. Page 21, line 446: "to decrease of the limit of", first "of" should be deleted.
3. References 36, 38, 39, 55 revise citations. Format should be consistent and article numbers/pages should be included.
4. In supplement: page 8, line 117, repeated word "Figure".
5. In supplement: page 14, line 247, "Figure 11Sb" should be "Figure S11b".
6. In supplement: page 16, line 291, "concentration the molecule" should be "concentration of the molecule".

Reviewer #3:

Remarks to the Author:

I had the pleasure to review this manuscript when it was submitted to ACS nano. I thought it should have been published then and I think it is even in a better shape now.

The introduction is beautifully written, clear and comprehensive. The authors have clearly put a lot of thoughts into this manuscript as experiments and controls have been crafted really well.

The topic of point of care sensing with nanopores is very timely and several papers have been published in this area, most of them in Nature Communications so I think this paper would be a great fit in this journal. Very few papers presented quantitative sensing with nanopores and an alternative strategy is a welcome addition and also the authors report an improved limit of detection compared to the state of the art which is remarkable.

My only reservation about the paper is that the detection strategy is fairly complex and ultimately it is an indirect readout of protein concentration. The authors only detect the presence of the DNA strand release by the UV treatment and to achieve this they need very small nanopores and high bandwidth amplifiers. Also, the expected range of concentration of the target molecule needs to be known beforehand to optimise the ratio junction strand to probe, this is far from ideal but the authors provide an exhaustive discussion in the manuscript about this matter.

As I also mentioned in my previous review, I think it is a beautiful work of the highest scientific standards... I personally doubt it could ever be used for point of care testing because of the complexity in the sample prep but I really hope the authors will prove me wrong!

I just have a couple of minor points for the authors to address"

the authors mention in their introduction "including the ability to tailor pore size to suit the target of interest on the fly"

This sentence needs clarification, how do you tailor the pore size on the fly? do the authors refer to their previous work about dielectric breakdown? if so they would need to add a citation.

I would also suggest the authors replace the bar plot in figure 4C with a dots plot so the reader can better understand the distribution of data.

I personally think figure S11 should be included in the main manuscript to complement figure 4 as panel B is not very informative

Response to the Referees' Comments

We would like to thank the reviewers for their helpful comments and suggestions. We have addressed the comments provided and feel that the paper is stronger for it. Point by point discussion of the concerns raised are below, along with appropriate callouts to edits in the paper. A version of the revised manuscript marked with the standard Track Changes feature of MS Word is also included, along with a clean revised manuscript.

Reviewer #1

Tessier, He and coworkers developed an assay for biomarker concentration quantification. The assay is based on the capture of antibodies on beads that have a releasable short oligo. Release is done by (photo-)cleavage of the oligo and then assembly of a DNA nanostructure into dumbbells that is then transferred to 3.2M LiCl salt concentrations to allow for detection in solid-state nanopores. The authors translate their target protein signal into a DNA signal (junction strand) that then is detected. The concept of translating a protein into a DNA signal is not novel and was published before in Kong *et al.* *Adv. Funct. Mat.* 2019 (<http://dx.doi.org/10.1002/adfm.201807555>). A paper that curiously is not cited. So the key idea to use nanopore sensing for the detection of proteins after translating the signal into a nanopore signal is not novel. The authors use a different type of DNA nanostructure to detect the presence of the junction strand by designing 12 arm shooting start structures. The paper is interesting for the nanopore sensing community as it shows another approach how DNA self assembly can be used to enable sensitive and specific sensing. However, the paper does not solve any problem that is relevant outside of the nanopore community as all scientific questions (dynamic range, false positive background etc.) are actually CAUSED by the chosen nanopore sensing system. The paper could be published in a journal like Nanotechnology or ACS sensors.

We would like to thank the reviewer for the comments and the concise summary of our work. We apologize for the oversight in missing Kong *et al.* (2019)¹ and we have cited this article in the revised version [new ref#28]. Note that we already did cite previous studies by the group of Ulrich Keyser at Cambridge that reported strand displacement followed by detection of barcodes on a DNA carrier, so there was nothing "curious" about omitting this particular reference. While Kong *et al.* (2019) is a great proof-of-concept study for multiplexed detection, they did not quantitate their target, nor did they measure from biofluids, and their targets are at a much higher concentration¹ (e.g. thrombin at 60 nM, ATP at 300 μ M). We agree that others have previously used DNA as labels for protein detection using different detection methodologies, and we also do not dispute that our manuscript is not the first to report a translation of signal for nanopore sensing. In fact, some of our team previously demonstrated ATP detection with a shape changing aptamer [Beamish *et al.* *ACS Sensors* 2017].

In this work, however, we present the first nanopore bead-based immunoassay scheme, which can be adapted for quantification of any target protein from serum samples down to the fM range. As mentioned by Reviewer 3, very few papers have reported protein quantification using solid-state nanopores. Here, we show significant improvements in sensing performances, as described in our initial cover letter (now reproduced at the end of this document). The revised manuscript further increase the sensitivity by >1,000x compared to previous studies quantifying

proteins from serum. We have included our updated Figure 4 below. Our revised nanopore assay is now capable of reaching the high fM range with a single pore. We feel that this is an exciting achievement that will be of interest to a wide audience interested in development of digital assays, in single-molecule electrical detection, and in advanced application of solid-state nanopores as biosensors all of which warrants publication in a journal like Nature Communications.

Figure 4. TSH assay calibration curve and TSH serum sample concentration quantification using solid-state nanopore digital detection. (a) TSH calibration curve concentration, 0 (blank), 0.15 nM, 0.3 nM, 0.6 nM, and 1.2 nM, repeated on pore 1 (10 nm, magenta diamonds), pore 2 (11.5 nm, blue triangles), pore 3 (12 nm, cyan circles), and fractions interpolated from data in Figure 3d (grey squares). The error bars represent one standard deviation following equation 3. (b) 0.48 nM TSH spiked serum as measured by the assay on the same pores. (c) Gold nanoparticle (AuNP) amplification assay results for TSH spiked in serum at 0.8 pM, 1.6 pM, 3.2 pM, 6.4 pM, and 12.8 pM on a 12.2 nm pore (orange squares). (d) Fraction of dumbbell events as a function of the effective event count $N^* = N_{DB} + N_P/2$ detected for dumbbells and probes on pore 1. The colored bands represent one standard deviation and the dashed lines represent the values of N^* needed for a given dumbbell fraction f to have $\sigma_f = 0.1, 0.05, 0.025, 0.0175, 0.01$. The probes P-1 and P-2 are fixed at 20 nM for 1:1 assay and 15 nM for the amplification assay. Experiments are performed in 3.2 M LiCl in a mixture with 2 kbp linear dsDNA at 100 mV and an intra-crossings threshold of 2.5x dsDNA, all experiments are low-pass Bessel filtered at 200 kHz for analysis.

Comment 1.1: The authors nicely summarise the challenge of detecting low concentration targets on page 4, lines 74-77: "This is due in part to the affinity of antibodies, since the dissociation constants for the available antibodies are typically larger than the clinically relevant concentration of the target in blood samples, and in part to the inadequate SNR of standard optical readouts." The authors address neither of these key issues on their work. They rely on

antibody binding to beads, introduce a nucleotide label that needs to be detected by a complicated process with nanopores. It is completely unclear why this could not be done with a quencher-dye, FRET or any other single-molecule optical assay as everything up and including the cleavage is not in any way relying on nanopore sensing. In case the authors want to make the paper relevant for the sensing community they need to compare their readouts with ELISA assays or the optical assays they are criticising. None of these things are presented in the manuscript.

The reviewer is correct to note that the upstream preparation steps for this assay are compatible with other readout mechanisms. In fact, the bead-based assay is similar to what is done in the SiMoA system by Quanterix, that uses optical readout to detect beads with and without bound proteins. We disagree however that we do not “*address neither of [the] key issues [associated with detecting low concentration targets] in [our] work*”. First, using antibody-coated beads multiplies the on rate of the capture antibody and reduces the K_D (dissociation constant) proportionately, so that we are not limited by the dissociation constant of a single antibody or aptamer, which otherwise limits the sensitivity to a small range around the K_D value. This argument is already addressed in the main text, which we have slightly changed so that it is clearer, on page 4 where we now state that:

“Some of these schemes are based on the use of paramagnetic micron-sized beads decorated with hundreds of thousands of capture antibodies, [ref. 45 main text] effectively turning each bead into a capture antibody with a significantly higher on-rate than that of individual antibodies, while same off-rate is maintained. [ref. 46, 47 main text]”

Secondly, the nanopore single molecule electrical readout allows for the implementation of a digital counting scheme, which frees the measurement from the error associated with the noise in an analog detector. As David R. Walt (Harvard) puts it: “*it is much easier to measure the presence or absence of signal [i.e. digital detection] than to detect the absolute amount of signal [i.e. analog detection]*”.² The sensitivity of our digital nanopore approach is limited by the measurement time, which can be accelerated through parallelization. Sub-fM sensitivity is therefore achievable with this scheme using an array of pores and a high-affinity antibody pair for the target. In the idealized limit of high levels of parallelization, the enhancement factor of the K_D by the magnetic bead would ultimately become the limiting factor. This is also discussed in the manuscript in length in the introduction on page 4 and 5 and before the conclusion on page 23.

To help motivate nanopore electrical sensors as being a worthwhile alternative to ELISA assays and other optical schemes, we note that the (i) intrinsically single-molecule nature of the measurement allowing digital schemes; (ii) the purely electrical readout, which generally provide a greater propensity for miniaturization and cost reduction when scaled, and (iii) the error modes affecting nanopore readout are completely different from those that effect fluorescence measurements, giving the possibility of improved sensing performances. For example in a digital fluorescence assay such as is implemented by Quanterix (SiMoA) beads are limited to <1 target

per bead to stay in the digital range and must be spatially separated so as to not be within the same illumination volume, the nanopore readout does not impose this. This is only one example, and including a comprehensive list of the pros and cons of single-molecule electrical counting versus optical methods is outside the scope of our manuscript and more suited for a review article. We believe that the manuscript already provides, on pages 4 and 5, a description of the limitation of analog methods justifying the need to digital count molecules to improve analytical performance. We have nonetheless modified a sentence of page 5 to clarify what possible advantages nanopore electrical readout can have as an alternative approach to fluorescence readout.

"Nanopore sensors with their intrinsically single-molecule resolution and purely electrical readout, are viewed as an attractive alternative to optical detection schemes, that offer a demonstrated path toward miniaturization and portability. [ref. 50 main text]"

Additionally, we have included the following references^{3,4} in the introduction of the main text, to provide the reader with additional information on the advantages and disadvantages of various optical and electrical methods for the detection of TSH. The text added on page 6 is as follows:

"TSH is frequently used as a proof-of-concept target in model research assays and in the design of biosensors with optical, electrochemical, or electrical readouts [ref. 51, 52 main text], since its clinically relevant concentration covers a large range and it is considered a good test case assay for new technologies since existing ELISA tests for TSH are particularly performant."

Comment 1.2: The key figure of the paper 4b contains a single TSH concentration measured with three different nanopores with non (or barely) overlapping error bars. Before any publication the authors have to present data with at least two more concentrations from the serum sample. The data in Figure 4b is simply not enough.

To address this comment, we have repeated our regular assay and performed five additional measurements in serum ranging from 150 pM to 1.2 nM. The data are shown in Supplementary Fig. 14 in Supplementary Notes 4 and below. Additionally, the data for the amplification assay shown in Figure 4c, provide another set of five measurement in serum ranging from 0.8 pM to 1.2 pM, for a total of 10 additional concentrations measurements from serum. On the contrary to the reviewer's assertion, the results shown in Figure 4b are overlapping within one standard deviation, the spike recovery is roughly 120% of the actual value, which is acceptable based on the accepted criteria found in the scientific literature⁵ and the CLSI guideline EP05 for spike recovery studies. Furthermore, our measurements were repeated on three different pores, which show an example of the relatively low pore-to-pore variance of our digital detection scheme.

Supplementary Figure 14. Additional TSH measurements in human serum sample. a) TSH calibration curve concentration, 0 (blank), 0.15 nM, 0.3 nM, 0.6 nM, and 1.2 nM. b) Fraction of dumbbell events as a function of the effective event count $N^* = N_{DB} + N_P/2$ detected for dumbbells and probes. Experiments are performed in 4x diluted human serum and sensed using a 12 nm pore in 3.2 M LiCl at 100 mV. Intra-crossings threshold is set 2.5x dsDNA, all experiments are low-pass Bessel filtered at 200 kHz for analysis.

Comment 1.3: The authors have to show that their method outperforms other detection systems like ELISA when the SAME process is used to translate the signal.

This is a perplexing comment since it precludes optimizing assay steps and translation process for one readout scheme over another. Using the same process to translate the signal when the readout method is entirely different ignores significant optimization opportunities that we have applied here. Nevertheless, we have carefully considered the reviewer's suggestion and performed such measurements with a homebrew ELISA employing the same workflow and reagents as our nanopore assay to validate our method. To do so, we replaced the biotinylated junction strand with a commercial biotinylated-HRP. Compared to the detection range ($\sim 0.1 - 70$ pM) of the commercially available ELISA for TSH (ThermoFisher Scientific), our homebrew ELISA performed relatively well, allowing us to detect low, medium and high pM values that corresponded with our nanopore assay. These results are shown in Supplementary Fig. 15, and below. These data allow us to validate our assay components much like the gel readout in Supplementary Fig. 12 and 13c, but a direct comparison with the nanopore assay performance would not yield very valuable insights at this stage. While the homebrew ELISA assay shows greater variability and a higher noise floor (the nanopore digital assay sensitivity is relative to the probe concentration, see response to Comment 1.1), there is potentially room for optimization if a different process is used to translate the signal.

Supplementary Figure 15. Validation of assay components by comparison of signal in sample diluent and low-mid-high values in human serum at nanopore assay concentrations. Commercial biotinylated HRP was used to produce an optical readout, with same assay components and steps scaled down to volumes compatible with a 96 well plate and a 24 magnet base. The values presented have been corrected by subtracting the absorbance of the blank. Briefly, primary antibody coated magnetic beads were washed with sample diluent then combined with TSH in either sample diluent or 1 in 4 diluted human serum sample and allowed to incubate for 1.5 h at room temperature (RT) with gentle agitation. The beads were washed three times for 5, 10, then 15 minutes, then were incubated with the secondary antibody in sample diluent for 1 h at RT. Following 3 quick washes, the beads were incubated with biotinylated HRP (ThermoFisher Scientific, 432040) for 15 min at RT. Following a final 3 quick washes the presence of TSH was indicated by incubation with TMB ELISA Substrate (fast kinetics ab171524) for 17.5 min then stopped using 450 nm Stop Solution for TMB Substrate (Abcam, ab171529). The absorbance at 450 nm was read using a BioTek EPCO 2 spectrophotometer (Biotek, BTEPOCH2).

Comment 1.4: The authors claim that nanopores are more amenable to bedside/point-of-care applications. For the nanopore sensor this is true but do they suggest that they can perform their assay in Figure 1 near the bedside on a nanopore system they have to show this.

We respectfully disagree with the reviewer's assertion that it is necessary to demonstrate bedside use of this nanopore sensing system. It is of course far beyond the scope of this proof-of-concept work to build an instrument capable of sample-in, answer-out bedside use.

Digital Microfluidics (DMF) technologies are well suited for performing similar bead-based assays in a compact and automated instrument.⁶ This is exemplified by the Voltrax commercialized with Oxford Nanopore Technologies for automating their library prep for MinION sequencing. Portability of nanopore devices is mentioned in the revised manuscript on page 5, when addressing Comment 1.1

Comment 1.5: In the outlook the authors make the claim that this system can reach fM detection levels for proteins when this is clearly limited by the antibody affinities as they

discussed in point 1). They need to either qualify this claim or explain how they aim to overcome the issue of low affinity antibodies as the nanopore sensor does not solve the key issue of affinity.

As discussed in our response to Comment 1.1, one of the advantages of the bead-based approach is the enhanced dissociation constant, since a bead with N antibodies behaves as a single antibody with an *on*-rate N times higher than that of each individual antibody, but with the same overall *off*-rate. This has been shown in literature and demonstrated to work experimentally, for example by Quanterix.^{2,7,8} To avoid similar confusion from readers, we have now made this fact clearer in the manuscript. Page 4 now reads:

"Some of these schemes are based on the use of paramagnetic micron-sized beads decorated with hundreds of thousands of capture antibodies [ref. 45 main text], effectively turning each bead into a capture antibody with a significantly higher on-rate than that of individual antibodies, but with the same off-rate."

Furthermore, as discussed in the manuscript, the micron sized beads are decorated with hundreds of thousands of capture antibodies, increasing the effective on rate for target capture by a bead by the same factor while target proteins that dissociate will immediately be recaptured by another capture antibody in the vicinity, effectively turning the paramagnetic beads into capture antibodies with fM sensitivity. To support this claim, we have performed additional experiments using a detection complex of gold nanoparticles decorated with hundreds of DNA oligos as an amplification scheme to quantitatively detect TSH down to ~385 fM in serum samples. The data are shown in Figure 4c of the revised manuscript, and added the paragraph below to page 18 of the manuscript. **This is >1,000x more sensitive than previous nanopore studies.**

"To further improve the sensitivity of our digital immunoassay and lower the LoD for a fixed nanopore counting time, we implemented an amplification scheme to detect rTSH down to the high femtomolar range. To achieve a ~100x amplification, we replaced the biotinylated ssDNA junction strand with a detection complex composed of 30 nm gold nanoparticles decorated with detector antibodies and hundreds of junction strands following protocols by Mirkin & co. [ref. 45, 71 main text] This effectively translates one protein into hundreds of ssDNA junction strand instead of a single one as illustrated in Figure 4c. Amplification experiments were carried out using an identical assay workflow as before. This time, we measured concentrations of spiked rTSH in human serum samples at 0.8 pM, 1.6 pM, 3.2 pM, 6.4 pM, and 12.8 pM. Figure 4d shows the detected fraction of dumbbell formed of 7.5 ±0.9%, 12 ±1%, 16 ±2%, 24 ±2%, and 41 ±2%, with a blank serum measured at 2.9 ±0.6%. LoD for the particular parameters of this amplified digital assay is calculated to be 385 fM."

Reviewer #2

This work reported a digital nanopore counting approach to quantify the target protein concentration from complex biofluids. The general strategy relies on the magnetic bead-based

sandwich immunoassay to convert the protein numbers to an equivalent number of junction ssDNA strands, which is then quantified by using the shooting star probes to form a recognizable dumbbell-like DNA nanostructure. In my assessment, this is a well-designed study and is of significant interest to the research community to develop ultrasensitive immunosensors. Major concerns I have for this strategy are the limited dynamic range, the complexity of the assay, and the sensing time. Although the authors did acknowledge these issues in the manuscript in a conceptual framework, these issues should be addressed with experimental data. I would recommend a major revision of the current manuscript.

We thank the reviewer for acknowledging the significance of our work and the helpful comments. The revised manuscript now shows a >1,000x improvement in sensitivity compared to previous nanopore studies. While we certainly acknowledge that future work will need to tackle assay time and dynamic range, we have provided responses to these comments below.

Comment 2.1: Page 8, line 176 mentioned "improving structural stability and reducing clogging in the nanopore". Authors should elaborate on why the internal carbon spacer can help reduce clogging. It's best to show experimental data to support this claim.

The internal carbon hexa-ethyleneglycol spacer has been shown to add flexibility within an oligo,^{9,10} the 18-atom spacer used in this study is the longest spacer arm available that can be added to an oligo by IDT, the extension combined with the added flexibility helps facilitate the assembly of the star-shaped structures.

Clogging of nanopores are not straightforward to quantify as it is often dependent on the nanopore itself. The reviewer is correct that we do not have experimental data to draw direct conclusions on the relative likelihood of clogging between the stars with and without the internal spacers. However, the star-shaped molecules without carbon spacers were characterized in our previous work.¹¹ We rephrased the sentence to on page 9 as follow:

"we further modified the star nanostructures, extending one of the arms to form a linkage for the tail section of the probe, and added an internal carbon spacer in the middle of each star oligo's sequence to relax the otherwise highly charged and sterically stressed core of the 12-arm star structure and to help facilitate translocation through nanopores."

Comment 2.2: Figure 2b, what does the label "2x dsDNA" mean? Please clarify if the unit dsDNA refers to the current blockage of the linear 2 kbp dsDNAs. I suggest using a plain vanilla current level so that other researchers in the community can benchmark (even though the pore size variation could change this current level, but readers can at least have some idea if authors can show an example).

We apologize for the confusion caused by this normalization. Yes, the unit of dsDNA refer to the current blockage level of 2kbp dsDNA, all experiments were done with linear 2kbp DNA in the mixture, they were added as a molecular ruler to help reduce pore-to-pore and experiment-to-

experiment variance in current blockage level, to ensure consistency in the threshold-crossing algorithm. For this reason, we feel it is a much better unit to use. Nonetheless, we have now indicated the corresponding plots in nA in Supplementary Fig. 8 in the revised version, which we note depends on pore size, pore length, voltage and salt solution conductivity. We rephrased the sentence on page 9 as below:

"We normalized the nanopore current signal by the blockage produced by the unfolded translocation of 2 kbp linear dsDNA to remove the effects of any variations in pore geometry and operating conditions between experiments, thus facilitating comparison between experiments on different pores as detailed in Supplementary Notes 3."

Comment 2.3: Figure 2c&d, the maximum blockage depths reported are normalized by the blockage of 2 kbp dsDNA. Since the 2kbp events were collected, it will be better to add it to the scatter plot in Figure 2c and 2d. what is 2kbp events' distribution. If using 2kbp dsDNA to normalize, what exact value should be used, the peak or the mean? In addition, what is the blockage shape for the 2kbp ruler? I suggest the authors show a representative shape for the 2kbp ruler.

We thank the reviewer for the opportunity to clarify this. We used the peak value from the 2kbp blockage distribution for normalization. We have now added scatter plots and current traces in Supplementary Fig. 8 in Supplementary Notes 3, showing the entire data including 2kbp DNA events with the blockage levels shown in nA. Representative current traces of 2kbp dsDNA, shooting star, and dumbbell events are also shown in Supplementary Fig. 8.

Supplementary Figure 8. Scatter plot and projected histograms of dwell time distribution and maximum blockage distribution of a mixture of nanostructures (shooting star, dumbbell, and 2kbp dsDNA) and their representative current traces. The nanopore experiment is performed using an 12 nm pore, in 3.2 M LiCl, with an applied bias of 100 mV and the intra-crossings threshold set to 2.5x dsDNA.

Comment 2.4: Figure 2d, please clarify if the dashed line (annotated 'p') in Figure 2d represents a replot of Figure 2c. How is the dwell time defined for dumbbell structure? From point 1 to 4 in figure 2b?

Yes, the dashed lines in Figure 2d are replots of the fitted histogram from Figure 2c to illustrate the shift in translocation time and blockage level. "P" is for probes and "DB" for dumbbells as shown in Figure 2a and mentioned in the caption.^{12,13} The dwell time of a dumbbell event is defined as the total translocation time of the event, from the time the current first leaves the baseline, to the point just before it returns to baseline. It is measured independently from the threshold crossings used to identify dumbbell translocations from unbound probe translocations. To avoid further confusions, we have added the following sentence to the caption of Figure 2:

"The fit to the probe distribution (P, red dash line) is overlaid with the dumbbell distribution to facilitate comparison between the two populations."

Comment 2.5: Figure 3, ten cases of different junction strand to probe pair ratios (from 0.025:1 to 10:1) were performed. It will be more informative to readers if the authors plot all of the scattering plots and indicate the number of detected events for each case.

We agree that adding all the scatter plots will be informative. We have added the corresponding scatter plots in Supplementary Fig. 9 and indicated the exact number of detected events at each concentration. Please note that approximate number of detected events at each concentration, ca. 1,100 events, was already given in the figure caption.

Supplementary Figure 9. Scatter plots and histograms of maximum blockage versus dwell time (log scale) of dose response in Figure 3, of junction strand to shooting star ratio from 0.01:1 to 20:1, showing 1639, 1524, 1268, 1337, 917, 882, 964, 1019, 1270, and 853 single-molecule events, respectively. Dumbbell events (red) are separated from shooting star events (dark grey) using the threshold-crossing algorithm.

Comment 2.6: Regarding the model (Eq.1), What is the physics behind your model? Any references? I suggest the authors rationale why this model applies here.

The rationale for the model is provided in the paragraph that follows the equation on page 13-14. In short, when binding is irreversible and there are fewer junction strands than probe pairs, the reaction will continue until all are bound (so $f_{DB} = x$ when $x < 1$). When there are more

junction strands than probes some probes will be capped, and it follows that this probability will be in proportion to the ratio of linkers to stars, hence $f_{DB} = 1/x$ when $x > 1$).

Comment 2.7: Figure 4, please add the pores 1,2 and 3 size information to the caption.

Pore size information has been added in the caption.

Comment 2.8: Page 23, line 501 mentioned '*TSH levels in the samples used are below the lower limit of detection (LoD 0.18 fM), therefore undetectable in our nanopore assay*'. What is the limit of detection in the "*Nanopore Digital Immunoassay for TSH*" section? I suggest adding the LoD analysis data.

The LoD of 0.18 fM referred to our homebrew SiMoA (Quanterix) assay using the same beads and antibody pair as in the nanopore assay. Testing the unspiked serum sample on the SiMoA allowed us to confirm that TSH levels were lower than our LoD with the nanopore assay, so as to not affect the spike recovery tests. The reviewer make a good suggestion of adding LoD analysis to work. We initially refrained from doing it because we viewed our work as a proof-of-concept for a nanopore-based digital immunoassay scheme and as discussed in the manuscript the sensitivity is fundamentally limited by the number of counted molecules which depends on the sensing time.

Nevertheless, we have now added LoD analysis for the two assays we tested. the lower limit of detection of our assay, where 1 DNA label is used per protein target shown in Figure 4a,b is 20 pM, and the LoD for our amplification assay, where gold nanoparticles decorated with multiple DNA labels are used, is 385 fM. This information has been added to the paper on page 17 and 19.

Comment 2.9: In supplement: Figure S7c showed some faint bands above the purified dumbbell bright band, and those were also observed in Figure S6a (from ratio 0.01:1 to 2.5:1). What is the possible cause for these bands?

The faint bands in Supplementary Fig. 6 and 7 are likely a result of multimers forming, since it is possible to have three or more shooting stars bind to each other. Due to the design of the dumbbell structures, misalignment of the component strands could potentially lead to partial binding, then it is possible for an oligo from one shooting start molecule to hybridize to the region of misalignment on another shooting star molecule, and therefore forming multimers and showing higher bands in gel electrophoresis.¹⁴ One source of such misassemblies could be misalignments or missing of individual strands on the structure from mechanical damage during the purification process (pipetting, centrifugation, etc.).

Comment 2.10: Based on the result in figure 3d, one should add a particular probe concentration to the system for the optimal range of junction strand to probe ratio. In a system with an unknown TSH concentration, how can one determine the probe concentration in the reaction to prevent junction to strand ratios higher than 1? Of course, as the author mentioned, testing the sample against different probe concentrations could, in principle, provide a 5-log dynamic range.

Nevertheless, considering the complexity of this strategy, authors should comment on the cost (if run parallelly) or the time to detect (if run sequentially). What are the key benefits of such a system as compared to other commercialized digital systems (e.g., Quanterix system). I would think this is the major drawback of the proposed digital assay towards real-world applications. These issues should be addressed with experimental data.

The reviewer raises a good point about the dynamic range of our assay, which we already discuss on page 15 of our manuscript. It is hazardous to comment on cost, since the cost incurred for this proof-of-concept study would be very different from a commercial test and is unlikely to be enlightening in terms of the long-term prospects of this type of test. This is not expected to be a limiting factor if a sample is split and measured in parallel.

The <2 logs of dynamic range for our nanopore assay for a fixed probe concentration is also currently limited by the relatively high limit of blank (LoB of 1-2%), which can most likely be improved by an order of magnitude with improvement with the design of our DNA nanostructures to reach 2.5-3 logs dynamic range for a single probe concentration. This limit of blank being so high was a result of non-optimal probe design on our part, which we discuss in detail on page 15 and 17. For comparison, the commercial SiMoA system from Quanterix has approximately 2.5 logs of usable digital immunoassay range available. Improvement to the DNA nanostructure design to improve false positive rates and reduce assay time is the subject of future work.

Typos and format

1. Page 9, line 189: the dumbbells translocate is in Figure 2b, not "Figure 2d".
2. Page 21, line 446: "to decrease of the limit of", first "of" should be deleted.
3. References 36, 38, 39, 55 revise citations. Format should be consistent and article numbers/pages should be included.
4. In supplement: page 8, line 117, repeated word "Figure".
5. In supplement: page 14, line 247, "Figure 11Sb" should be "Figure S11b".
6. In supplement: page 16, line 291, "concentration the molecule" should be "concentration of the molecule".

We thank the reviewer for the corrections, we have fixed them in the revised manuscript.

Reviewer #3

I had the pleasure to review this manuscript when it was submitted to ACS nano. I thought it should have been published then and I think it is even in a better shape now.

The introduction is beautifully written, clear and comprehensive. the authors have clearly put a lot of thoughts into this manuscript as experiments and controls have been crafted really well.

We appreciate the reviewer's very insightful comments and the acknowledgement on the quality of our work. We have provided our responses to these points below and revised the manuscript accordingly.

Comment 3.1: The topic of point of care sensing with nanopores is very timely and several papers have been published in this area, most of them in Nature Communications so i think this paper would be a great fit in this journal. Very few papers presented quantitative sensing with nanopores and an alternative strategy is a welcome addition and also the authors report an improved limit of detection compared to the state of the art which is remarkable.

We thank the reviewer for acknowledging our work. We agree that this is one of the few works that show nanopore quantification of proteins found in serum samples, and we believe this is the first work to show bead-based immunoassay integrated into a nanopore sensing platform. The reviewer should be pleased that the revised manuscript further improves the limit of detection (LoD) by two orders of magnitude, with a LoD of 385 fM. This work is now providing a >1,000x improvement in sensitivity compared to previous nanopore studies sensing proteins from serum (see Figure 4).

Comment 3.2: My only reservation about the paper is that the detection strategy is fairly complex and ultimately it is an indirect readout of protein concentration. The authors only detect the presence of the DNA strand release by the UV treatment and to achieve this they need very small nanopores and high bandwidth amplifiers. Also, the expected range of concentration of the target molecule needs to be known beforehand to optimise the ratio junction strand to probe, this is far from ideal but the authors provide an exhaustive discussion in the manuscript about this matter.

The use of paramagnetic bead-based assay is our approach to solving the challenges often associated with direct protein sensing with nanopores, such as clogging, fast passage times/missed events, lack of specificity from similar looking translocation signatures, and lack of required sensitivities, etc. The bead-based assay allows the measurement of any target protein with very high sensitivities and specificity without needing to compromise on optimal nanopore sensing conditions (also see answer to Comment 1.1 about not being limited by the K_D of an Ab); the indirect readout provides very low pore-to-pore variance; and the use of proxy labels avoids the clogging that usually comes with protein sensing in biofluids. Regarding the pore size constraint, our previous study¹¹ has shown that we could reliably detect the star-shaped DNA labels in a wide range of pore sizes, from 4 to 13 nm. We therefore do not require very precise or very small pores. As pointed out by the reviewer, these improvements come at the cost of a more complex sample preparation – though such assay workflow can be automated on DMF system for example. We discuss the dynamic range question in our answer to Comment 2.10 which we are hoping to address in a future study.

Comment 3.3: As I also mentioned in my previous review, i think it is a beautiful work of the highest scientific standards... I personally doubt it could ever be used for point of care testing because of the complexity in the sample prep but I really hope the authors will prove me wrong!

The reviewer has a good point: the sample prep requires multiple steps in the current assay design. However, we believe that all sample prep steps can be automated, and we hope to integrate the nanopore assay with a digital microfluidic system going forward. It is worth noting that the sample prep is no more complex than that required for the Quanterix system, which has in fact been fully automated (albeit in a fairly bulky and expensive form factor).

Comment 3.4: I just have a couple of minor points for the authors to address: the authors mention in their introduction "*including the ability to tailor pore size to suit the target of interest on the fly*" This sentence needs clarification, how do you tailor the pore size on the fly? do they authors refer to their previous work about dielectric breakdown? if so they would need to add a citation.

We thank the reviewer for pointing this out. Yes, our nanopores are fabricated using controlled breakdown (CBD) and our systems are capable of fine-tuning pore size on handheld device. Our most recent work that demonstrates this capability is cited in the article.

Comment 3.5: I would also suggest the authors replace the bar plot in figure 4C with a dots plot so the reader can better understand the distribution of data.

We have reformatted Figure 4 to include new data from the amplification assay that employs gold nanoparticles decorated with secondary antibodies and hundreds of junction strands as opposed to one DNA label per target protein (see response to Comment 1.1).

Comment 3.6: I personally think figure s11 should be included in the main manuscript to complement figure 4 as panel B is not very informative.

Following the reviewer suggestion, we have removed Figure 4b from the manuscript. Supplementary Fig. 11 is an informative comparison of a few different counting methods, however, we are worried it would distract the readers from the main message in the main manuscript.

Editor's Comments

As you will see from the reports copied below, the reviewers raise important concerns. We find that these concerns limit the strength of the study, and therefore we ask you to address them with additional work. Without substantial revisions, we will be unlikely to send the paper back to review. In particular, we ask for the comparison to ELISA and optical readouts to justify your method, as suggested by Reviewer #1, and comparison to commercial systems and explanation of the application, as suggested by Reviewer #2.

We thank the editor for the feedback. We have addressed the reviewers' comments point-by-point in this rebuttal. In response to Reviewer #1's suggestion of adding a comparison to ELISA and optical readouts. We have provided detailed comparisons between ELISA and our assay scheme in our response to Comments 1.1 and 1.3. Moreover, following the reviewer's suggestions, we performed a homebrew ELISA, using the same workflow and reagents as our nanopore assay as validation and comparison in our response to Comment 1.3. Regarding Reviewer #2's suggestion of a comparison to commercial systems and explanation of the application, we have provided a discussion in response to Comment 2.10, where we compared our system to the SiMoA in terms of limit of blank and dynamic range, as well as potential strategies for improvements. Furthermore, we included a brief comparison of our nanopore amplification assay (0.4 – 12 pM) to commercial ELISA (0.1 – 70 pM) in our response to Comment 1.3, where we noted that the sensitivity of our digital nanopore approach is limited by the measurement time, which can be accelerated through parallelization. Sub-fM sensitivity is therefore achievable with this scheme using an array of pores and a high-affinity antibody pair for the target. In the idealized limit of high levels of parallelization, the enhancement factor of the K_D by the magnetic bead is ultimately the limiting factor. In addition, we implemented an amplification assay with the use of gold nanoparticle complex, this allowed us to improve our sensitivity by >1,000x compared to previous nanopore studies quantifying proteins from serum.

References

1. Kong, J., Zhu, J., Chen, K. & Keyser, U. F. Specific Biosensing Using DNA Aptamers and Nanopores. *Adv. Funct. Mater.* **29**, 1807555 (2019).
2. Walt, D. R. Optical methods for single molecule detection and analysis. *Anal. Chem.* **85**, 1258–1263 (2013).
3. Park, S. *et al.* Combined Signal Amplification Using a Propagating Cascade Reaction and a Redox Cycling Reaction for Sensitive Thyroid-Stimulating Hormone Detection. *Anal. Chem.* **91**, 7894–7901 (2019).
4. Gutiérrez-Sanz, Ó., Andoy, N. M., Filipiak, M. S., Hausteine, N. & Tarasov, A. Direct, Label-Free, and Rapid Transistor-Based Immunodetection in Whole Serum. *ACS Sensors* **2**, 1278–1286 (2017).
5. Wu, D., Katilius, E., Olivas, E., Dumont Milutinovic, M. & Walt, D. R. Incorporation of Slow Off-Rate Modified Aptamers Reagents in Single Molecule Array Assays for Cytokine Detection with Ultrahigh Sensitivity. *Anal. Chem.* **88**, 8385–8389 (2016).

6. Dixon, C., Lamanna, J. & Wheeler, A. R. Direct loading of blood for plasma separation and diagnostic assays on a digital microfluidic device. *Lab Chip* **20**, 1845–1855 (2020).
7. Rissin, D. M. *et al.* Single-molecule enzyme-linked immunosorbent assay detects serum proteins at subfemtomolar concentrations. *Nat. Biotechnol.* **28**, 595–599 (2010).
8. Wilson, D. H. *et al.* The Simoa HD-1 Analyzer: A Novel Fully Automated Digital Immunoassay Analyzer with Single-Molecule Sensitivity and Multiplexing. *J. Lab. Autom.* **21**, 533–547 (2016).
9. Salunkhe, M., Wu, T. & Letsinger, R. L. Control of Folding and Binding of Oligonucleotides by Use of a Nonnucleotide Linker. *J. Am. Chem. Soc.* **114**, 8768–8772 (1992).
10. Dolinnaya, N. G. *et al.* Oligonucleotide circularization by template-directed chemical ligation. *Nucleic Acids Res.* **21**, 5403–5407 (1993).
11. He, L., Karau, P. & Tabard-Cossa, V. Fast capture and multiplexed detection of short multi-arm DNA stars in solid-state nanopores. *Nanoscale* **11**, 16342–16350 (2019).
12. Briggs, K. Solid-State Nanopores: Fabrication, Application, and Analysis. (PhD thesis, University of Ottawa, 2018).
13. Forstater, J. H. *et al.* MOSAIC: A modular single-molecule analysis interface for decoding multistate nanopore data. *Anal. Chem.* **88**, 11900–11907 (2016).
14. Seeman, N. C. Nucleic acid junctions and lattices. *J. Theor. Biol.* **99**, 237–247 (1982).

Cover Letter

Our initial cover letter is attached below:

10 November 2020

Subject: Submission to Nature Communications

Dear Editors,

Please find enclosed a manuscript to be submitted for publication to Nature Communications as an Article. The work is entitled "*Digital Immunoassay for Biomarker Concentration Quantification using Solid-State Nanopores*" and is authored by Daniel Tessier, Liqun He, Kyle Briggs, Matthaios Tsangaris, Martin Charron, and Vincent Tabard-Cossa.

In the last few years, motivated by the need for novel digital diagnostic technologies, the biosensing field has seen important advances on nanopore detection of proteins in clinically relevant samples. Some of these studies have been published by your journal, in particular Sze *et al.* Nat Commun 2017 [i], Cai *et al.* Nat Commun 2019 [ii], and Ravendraan *et al.* Nat Commun 2020 [iii]. **The present manuscript presents significant advances in this active area of nanopore-based digital immunoassays that clearly sets this work apart from previous studies.**

Our manuscript builds on our previous publications from 2019 and 2020, in *Nanoscale* [1,2] that fully characterizes the capture and translocation of specific DNA nanostructures, in *Analytical Chemistry* [3] that introduces the method of controlled counting to precisely measure the

concentration of a target, and in *Nature Protocols* [4] that presents our latest tools to fabricate solid-state nanopores by the controlled breakdown method which we invented. Here, we develop a versatile **nanopore-based digital immunoassay scheme capable of reliably measuring the concentration of a specific target protein in serum**. We employ a **magnetic bead-based sandwich immunoassay upstream of nanopore-based electrical sensing** which has clear potential for ultrasensitivity. Our scheme converts the target protein into proxy labels configured as **12-arm star DNA nanostructures to quantify its presence ("1") or absence ("0")**, providing a strong SNR and high accuracy to quantify the biomarker concentration. **These proxy labels allow for an automatable signal analysis method to digitally classify electrical signatures** from the different translocating DNA nanostructures, eliminating manual classification and user biases.

These advances combined allow us to overcome what we believe are **the two main challenges associated with the use of solid-state nanopores for reliable, ultra-sensitive protein concentration quantification from complex biological samples**: first, the specificity for the target signal in the presence of a much higher concentration of background molecules; and second, the consistency of downstream nanopore electrical detection between pores of varying geometry. **We use our method to quantify, with high precision, the concentration of thyroid-stimulating hormone (TSH) from serum in the picomolar to nanomolar range.**

We believe that this work represents a major advance in solid-state nanopore biosensing that will be of interest to the many researchers in this field that have been awaiting exciting results on the development of promising diagnostic applications with solid-state nanopores. **Compared to previous nanopore studies [i-v] in the field referenced below, the digital scheme presented here improves on:**

- **Sensitivity:** We demonstrate a 10 – 100x improvement in sensitivity compared to the previous nanopore studies sensing proteins from serum. *[note: the revised manuscript now demonstrate 10,000 – 1,000x improvements]*
- **Dynamic Range:** Our assay shows better dynamic range and precision, we can precisely quantify the concentration of a protein target over ~2-logs compared ~1-log in previous studies.
- **Specificity/Selectivity: (a)** Our magnetic bead-based digital immunoassay scheme allows us to use high affinity antibodies to capture protein targets, whereas DNA nanocarrier schemes are in practice limited to lower affinity aptamers. **(b)** Protein capture can be done under optimized conditions to efficiently capture all targets, whereas DNA nanocarrier schemes require mixing serum with sensing solution, which is neither optimized for highest affinity binding to targets nor for best nanopore sensing conditions. **(c)** The use of magnetic beads allow washing away background molecules, and non-specifically bound objects, leaving only molecules of interest to translocate through the pore. This significantly increases selectivity and is ideal for serum samples
- **Electrical SNR:** We show a SNR of 20 (at 200 kHz), >4x compared to previous studies, while using a 2-20x higher recording bandwidth to better resolve molecules.

- **Robustness:** Our rigid DNA nanostructures result in clear, consistent, and easily identifiable translocation signatures, with great pore-to-pore consistency, eliminating false positives from ambiguous molecular folding.
- **Precision, Reliability and Reproducibility:** We recorded 10 – 100x more single molecule events compared to previous studies, achieving high precision calibrations by collecting thousands of single-molecule events in minutes. In total, the work is comprised of $>10^5$ single-molecule events acquired on 13 nanopores, surpassing the standard of most articles.

Looking forward, our work also shows the potential for:

- **aM to fM sensitivity limits:** Using antibody-coated beads multiplies the on rate of the capture antibody and reduces the K_D proportionately, so that we are not limited by the dissociation constant of a single antibody or aptamer, which otherwise limits the sensitivity to a small range around the K_D value. The sensitivity of our approach is currently limited only by measurement time. By parallelizing the pore detection, we could reach fM sensitivity for proteins and aM for nucleic acid targets. In contrast, schemes that employ a single receptor to bind a target are limited to a small target concentration range around the K_D for the receptor, making them incompatible with low-abundance biomarker detection.
- **Scalability:** Our solid-state nanopores are fabricated on a substrate that can be scaled for manufacturing and commercialization. Nanopipettes, used so far in previous digital assay [i-iii] studies are inherently limited to research settings due to their form factors and their manual and delicate fabrication procedure.

We believe these facts to be accurate and hope that putting our results in clear perspective with prior nanopore studies will help you reach a favourable decision to send this work for peer review.

Please feel free to contact me if you require more information.

Sincerely,

Vincent Tabard-Cossa
Associate Professor,
Member, College of the Royal Society of Canada
Department of Physics
School of Electrical Engineering and Computer Science
University of Ottawa, Ottawa, ON, Canada
tcossa@uOttawa.ca

References:

This work is built on our recent previous work:

- 1) He, L., Karau, P., Tabard-Cossa V. Fast capture and multiplexed detection of short multi-arm

DNA stars in solid-state nanopores. *Nanoscale*. **11**, 16342-16350 (2019).

- 2) Beamish, E., Tabard-Cossa V. & Godin, M. Digital counting of nucleic acid targets using solid-state nanopores. *Nanoscale*. **12**, 17833–17840 (2020)
- 3) Charron, M., Briggs, K., King, S., Waugh, M., Tabard-Cossa, V. Precise DNA Concentration Measurements with Nanopores by Controlled Counting. *Analytical Chemistry*. **91**, 12228-12237 (2019).
- 4) Waugh, M., ..., Tabard-Cossa, V. Solid-state nanopore fabrication by automated controlled breakdown. *Nature Protocols* **15**, 122–143 (2020).

and compares favourably to previous nanopore studies detecting proteins from serum:

- i) Sze, J. Y. Y., Ivanov, A. P., Cass, A. E. G. & Edel, J. B. Single molecule multiplexed nanopore protein screening in human serum using aptamer modified DNA carriers. *Nat. Commun.* **8**, 1552 (2017).
- ii) Cai, S.; Sze, J. Y. Y.; Ivanov, A. P.; Edel, J. B. Small Molecule Electro-Optical Binding Assay Using Nanopores. *Nat. Commun.* **10**, 1797 (2019).
- iii) Raveendran, M.; Lee, A. J.; Sharma, R.; Wälti, C.; Actis, P. Rational Design of DNA Nanostructures for Single Molecule Biosensing. *Nat. Commun.* **11**, 4384 (2020).
- iv) Chuah, K.; Wu, Y.; Vivekchand, S. R. C.; Gaus, K.; Reece, P. J.; Micolich, A. P.; Gooding, J. J. Nanopore Blockade Sensors for Ultrasensitive Detection of Proteins in Complex Biological Samples. *Nat. Commun.* **10**, 2109 (2019).

Reviewers' Comments:

Reviewer #1:

Remarks to the Author:

The authors provided more data and improved their manuscript. The overall assessment of the work - however - remains that the work is introducing an extremely complex sample preparation. The authors provided more data and the comparison with ELISA is welcome. The discussion of the limitations and weaknesses are as well clearer now. The reply to the reviewers and the additional supplementary figures will help readers to judge the work.

In conclusion the paper can now be published.

Reviewer #2:

Remarks to the Author:

While major concerns I pointed out last time (the limited dynamic range, the complexity of the assay, and the sensing time) were not addressed directly, I think this revision is much more improved than the last submission. I am in favor of its publication.

Reviewer #3:

Remarks to the Author:

The authors have fully addressed all my comments and I think the manuscript should be accepted for publication!